# USP36 stabilizes nucleolar Snail1 to promote ribosome biogenesis and cancer cell survival upon ribotoxic stress

Kewei Qin[1,5], Shuhan Yu[1,5], Yang Liu[1], Rongtian Guo[1], Shiya Guo[1], Junjie Fei[1], Yuemeng Wang[1], Kaiyuan Jia[1], Zhiqiang Xu[2], Hu Chen ®[1,3], Fengtian Li ®[1], Mengmeng Niu ®[1], Mu-Shui Dai ®[4], Lunzhi Dai ®[2], Yang Cao ®[1], Yujun Zhang[1], Zhi-Xiong Jim Xiao ®[1,2] ✉ & Yong Yi ®[1] ✉

Tumor growth requires elevated ribosome biogenesis. Targeting ribosomes is an important strategy for cancer therapy. The ribosome inhibitor, homo-harringtonine (HHT), is used for the clinical treatment of leukemia, yet it is ineffective for the treatment of solid tumors, the reasons for which remain unclear. Here we show that Snail1, a key factor in the regulation of epithelial-to-mesenchymal transition, plays a pivotal role in cellular surveillance response upon ribotoxic stress. Mechanistically, ribotoxic stress activates the JNK-USP36 signaling to stabilize Snail1 in the nucleolus, which facilitates ribosome biogenesis and tumor cell survival. Furthermore, we show that HHT activates the JNK-USP36-Snail1 axis in solid tumor cells, but not in leukemia cells, resulting in solid tumor cell resistance to HHT. Importantly, a combination of HHT with the inhibition of the JNK-USP36-Snail1 axis synergistically inhibits solid tumor growth. Together, this study provides a rationale for targeting the JNK-USP36-Snail1 axis in ribosome inhibition-based solid tumor therapy.

Ribosomes are essential for protein production, and thus for cell proliferation and survival. Ribosome biogenesis is initiated in the nucleolus including the synthesis and processing of ribosomal RNAs (rRNAs), assembly of ribosomal proteins, transport to the cytoplasm, and association of ribosomal subunits for protein synthesis. The dys-regulation of ribosome biogenesis is often associated with cancer, aging, and age-related degenerative diseases[1].

Tumor growth requires elevated ribosome functions for rapid protein synthesis, which is often resulted from increased ribosome biogenesis in the nucleoli, representing a specific hallmark of cancer cells[2]. Thus, inhibition of the ribosome function has been considered an important strategy for cancer therapy. Translation inhibitors (such as anisomycin, blasticidin, and cycloheximide), chemotherapeutics

(such as doxorubicin), ribotoxins (such as ricin, Shiga toxin, and α-sarcin), and UV radiation, can impair ribosome function to trigger the ribotoxic stress, which often leads to robust activation of p38 and JNK signaling to trigger cancer cell apoptosis[3].

Homoharringtonine (HHT), a plant alkaloid originally isolated from *Cephalotaxus harringtonia* several decades ago, has been shown to bind to the A-site cleft in the peptidyl transferase center of the ribosome to block protein synthesis[4]. Currently, HHT is the only ribosome inhibitor specifically used for the treatment of acute myeloid leukemia (AML), chronic myeloid leukemia (CML), and myelodys-plastic syndrome (MDS)[5]. HHT induces the rapid turnover of several key oncoproteins, including c-Myc and Mcl-1, and potently triggers apoptosis in leukemia cells[6,7]. However, it has been documented that

[1]Center of Growth, Metabolism and Aging, Key Laboratory of Bio-Resource and Eco-Environment of Ministry of Education, College of Life Sciences, Sichuan University, 610064 Chengdu, China. [2]State Key Laboratory of Biotherapy, West China Hospital, Sichuan University, 610041 Chengdu, China. [3]Department of Cardiothoracic Surgery, First Affiliated Hospital of Chengdu Medical College, 610500 Chengdu, China. [4]Department of Molecular & Medical Genetics, Oregon Health & Science University, Portland, OR, USA. [5]These authors contributed equally: Kewei Qin, Shuhan Yu. ✉e-mail: jimzx@scu.edu.cn; yy-yiyong@scu.edu.cn

HTT exhibits little anticancer activity on solid tumors[8], the reasons for which remain unclear.

Snail1 is a key transcription factor in the regulation of epithelial-to-mesenchymal transition (EMT), cell survival, metabolic reprogramming, and cancer stemness[9]. Snail1 protein is mainly localized in the nucleoplasm and represses Pol II-mediated transcription of *CDH1* to promote EMT. Snail1 can also function as a survival factor since high-level Snail1 expression leads to tumor resistance to many chemotherapeutic drugs[9]. Snail1 is a highly unstable protein and is degraded in both cytosol and nucleus. Several E3 ligases, including β-TrCP1, FBXW7, FBXL14, FBXL5, FBXO11, FBXO22, FBXO31, FBXO45, TRIM50, HECTD1, CHIP, PPIL2, SPSB3, and TRIM21, have been reported to promote Snail1 proteasome-mediated degradation[10–16]. Deubiquitinases (DUBs) also play an important role in the regulation of Snail1 protein stability, including USP3, USP11, USP13, DUB3, USP18, USP26, USP27X, USP37, USP47, OTUB1, and PSMD14, that have been shown to facilitate deubiquitination and stabilization of nuclear Snail1 in promoting cell proliferation, migration, and invasion[11,17,18]. USP29 has been shown to stabilize Snail1 in the nucleus to enhance the chemoresistance of lung cancer cells[19].

In this study, we demonstrate that ribotoxic stress promotes Snail1 accumulation in the nucleolus and facilitates ribosome biogenesis and cancer cell survival independent of EMT-regulating function. USP36 is transcriptionally upregulated through the JNK-HSF1 axis upon ribotoxic stress, and functions as a nucleolar deubiquitinase to stabilize Snail1 protein. A combination of HHT with inhibition of the JNK-USP36-Snail1 axis synergistically inhibits solid tumor cell viability in vitro and tumor growth in vivo.

## Results

### Ribotoxic stress induces Snail1 nucleolar accumulation

Tumor growth requires elevated ribosome biogenesis in the nucleoli essential for rapid protein synthesis, representing a hallmark of cancer cells[2]. Snail1, primarily localized in the nucleoplasm, is a key transcription factor regulating the expression of a set of genes important for epithelial-to-mesenchymal transition (EMT), stemness, drug resistance, and inflammation[9,20,21]. However, whether Snail1 can be localized in the nucleolus to regulate ribosome biogenesis remains unknown. To explore these issues, we first performed immunofluorescence assays to examine the subcellular localization of the Snail1 protein. As shown in Fig. 1a and Supplementary Table 1, the majority of Snail1 was localized in the nucleoplasm in human triple-negative breast cancer HCC1806 cells, notably, a small percentage (approximately 4%) of HCC1806 cells showed Snail1 co-localized with the nucleolar marker B23, in which about ~18% of total nuclear Snail1 protein was attributable to the nucleolus (Supplementary Table 2). Interestingly, the co-localization of Snail1 and B23 was also observed in the CDX (cell-derived xenograft) tumors derived from human lung cancer A549 cells and in the clinical breast tumor samples (Fig. 1a), suggesting that Snail1 could be localized in the nucleolus.

Ribosome biogenesis is initiated in the nucleolus including the transcription and processing of ribosomal RNAs (rRNAs) and the assembly of ribosomal proteins. It has been well documented that inhibition of ribosomes often leads to ribotoxic stress and the obstruction to translation[3]. HHT (homoharringtonine), a ribotoxic stress inducer, has been widely used in the treatment of acute non-lymphocytic leukemia[5]. We therefore examined whether HHT-induced ribotoxic stress can impact Snail1 nucleolar localization. As shown in Fig. 1b, c, HHT treatment led to Snai11 accumulation in the nucleoli in HCC1806 cells, as evidenced by the co-localization of Snail1 and B23. Similar observations were obtained in triple-negative breast cancer SUM159 and Hs 578T cells as well as in non-small cell lung cancer A549 cells (Fig. 1d, e). By contrast, under the same experimental settings, either the mTOR inhibitor rapamycin, a Pol I inhibitor CX-5461[22], or an ER stress inducer tunicamycin, failed to do so (Fig. 1b, c). Notably,

similar to HHT, other ribotoxic stress inducers, including anisomycin, puromycin, G418, and blasticidin[23–25], also led to a dramatic increase of Snail1 nucleolar accumulation (Fig. 1f, g). Furthermore, the cellular fractionation assay showed that while basal levels of Snail1 were detected primarily in the nucleoplasm, HHT robustly induced Snail1 nucleolar accumulation (Fig. 1h). Together, these results indicate that ribotoxic stress can lead to Snail1 accumulation in nucleoli.

### Ribotoxic stress induces nucleolar Snail1 accumulation via upregulation of USP36

We next investigated the molecular basis by which ribotoxic stress induces Snail1 accumulation in nucleoli. As shown in Fig. 2a and Supplementary Fig. S1a, HHT significantly increased Snail1 protein half-life while Snail1 mRNA levels were mildly elevated. Since Snail1 protein stability can be regulated by the balance of ubiquitination and deubiquitination, it is conceivable that altered expression of ubiquitin E3 ligases or deubiquitinases specific for Snail1 can affect Snaill1 protein stability. Indeed, nuclear Snail1 protein stability can be regulated by several ubiquitin E3 ligases and deubiquitinases to impact EMT[26]. However, how nucleolar Snail1 protein stability is regulated remains unknown. Therefore, we aimed to identify nucleolar deubiquitinase impacting Snail1. With this regard, we analyzed the Human Protein Atlas Database (https://www.proteinatlas.org/) and found that there are nine deubiquitinases, which can be localized in the nucleoli as shown by the immunofluorescence assay but there lacks subsequent verification (Supplementary Table 3). We then ectopically expressed each of the nine Flag-tagged deubiquitinases in HEK-293 cells stably expressing HA-Snail1. As shown in Fig. 2b and Supplementary Fig. S1b, only USP36 (a confirmed deubiquitinase exclusively located in the nucleolus[27]) and ATXN3, but not the other seven nucleolar deubiquitinases, markedly upregulated HA-Snail1 protein expression, comparable to the MG132-induced stabilization of HA-Snail1 (MG132 is proteasome inhibitor and thus blocks all proteasome-mediated degradation). Furthermore, ribotoxic stress inducers, including HHT, puromycin, anisomycin, and blasticidin, significantly upregulated USP36 protein expression as well as Snail1 expression (Fig. 2c, d). Notably, ectopic expression of USP36, but not USP36^C131A defective in deubiquitinase activity[27], upregulated endogenous Snail1 protein expression in both SUM159 and HCC1806 cells (Fig. 2e). Ectopic expression of USP36 increased Snail1 protein expression in a dose-dependent manner (Supplementary Fig. S1c). Conversely, the knockdown of USP36 led to the downregulation of Snail1 (Supplementary Fig. S1d), which was completely rescued by the restoration of USP36, but not USP36^C131A (Fig. 2f). Moreover, the silencing of USP36-mediated downregulation of Snail1 could also be totally rescued by proteasome inhibitor MG132 (Supplementary Fig. S1e). Importantly, ectopic expression of USP36, but not USP36^C131A, upregulated endogenous Snail1 accumulation in the nucleoli (Fig. 2g). Silencing of USP36 significantly inhibited HHT-induced Snail1 accumulation in the nucleoli (Fig. 2h). Together, these results indicate that USP36 plays a critical role in ribotoxic stress-induced nucleolar Snail1 accumulation.

### USP36 is a deubiqutinase of nucleolar Snail1

USP36 is exclusively located in nucleoli[27–29]. We therefore examined the co-localization of Snail1 and USP36 by immunofluorescence assays. As shown in Fig. 2h and Supplementary Fig. S1f, while endogenous Snail1 was detected primarily in the nucleoplasm, HHT robustly induced co-localization of Snail1 and USP36 in nucleoli. To investigate whether USP36 is a deubiquitinase of Snail1 protein, we examined the effects of USP36 on Snail1 protein stability. Stable protein complexes between USP36 and Snail1 were readily detected in the nucleoli even in the presence of 500 mM NaCl (Fig. 3a). Ectopic expression of USP36, but not USP36^C131A, significantly upregulated Snail1 protein half-life (Fig. 3b). Silencing of USP36 markedly reduced Snail1 protein

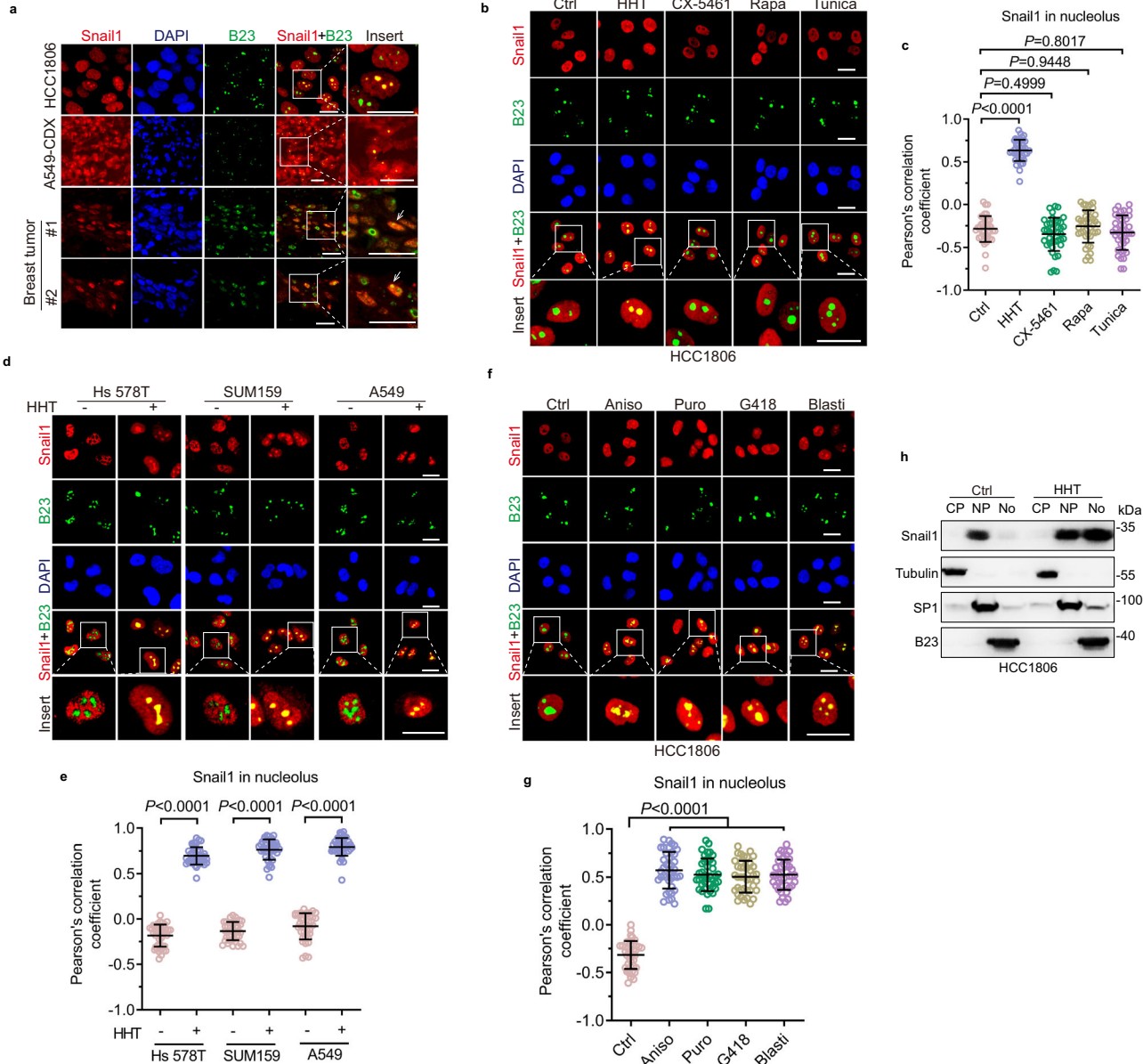

**Fig. 1 | Ribotoxic stress promotes Snail1 nucleolar accumulation.**
**a** Immunofluorescence staining assays were performed to examine Snail1 or nucleolus marker B23 in either fixed HCC1806 cells, frozen sections of A549 cell-derived xenograft tumor (A549-CDX), or paraffin sections of two clinical breast tumor samples. Notice that Snail1 was accumulated in the nucleolus in a few cells. **b**, **c** HCC1806 cells were treated with ribosome inhibitor HHT (20 ng/mL and hereafter), a Pol I inhibitor CX-5461 (200 nM), mTOR inhibitor rapamycin (Rapa, 20 nM), or ER stress inducer Tunicamycin (Tunica, 2 μg/mL) for 24 h. Cells were subjected to immunofluorescence staining analyses (**b**). The co-localization between Snail1 and B23 (as analyzed by Pearson's correlation coefficient[53,55]) was quantified and statistically analyzed (**c**). **d**, **e** Hs 578T, SUM159, or A549 cells were treated with or without HHT for 24 h. Cells were subjected to immunofluorescence staining analyses (**d**). The co-localization between Snail1 and B23 was quantified and

statistically analyzed (**e**). **f**, **g** HCC1806 cells were treated with a ribotoxic inducer anisomycin (Aniso, 50 ng/mL), puromycin (Puro, 200 ng/mL), G418 (1 μg/mL), or blasticidin (Blasti, 2 μg/mL) for 24 h. Cells were subjected to immunofluorescence staining analyses (**f**). The co-localization between Snail1 and B23 was quantified and statistically analyzed (**g**). **h** HCC1806 cells were treated with or without HHT for 24 h followed by cell fractionation and western blot analyses. CF Cellular fraction, CP Cytoplasm, NP Nucleoplasm, No Nucleolus. This experiment has been repeated for three times with similar results. Quantification of the co-localization between Snail1 and B23 using Pearson's correlation coefficient (**c**, **e**, **g**). 40 cells derived from three independent experiments were randomly chosen and subjected to quantification analyses. Data were presented as mean ± SD. Comparisons were performed with unpaired two-tailed Student's *t* test. Scale bar, 25 μm.

half-life (Fig. 3c). In addition, USP36, but not USP36^C131A, could remove the polyubiquitin chain of Snail1 (Fig. 3d). Consistently, the knockdown of USP36 remarkably facilitated Snail1 protein polyubiquitination (Fig. 3e). Notably, USP36 could remove the K48-mediated polyubiquitination of Snail1, but not the K63-mediated polyubiquitination (Fig. 3f).

Next, we aimed to identify specific amino acid residues on the Snail1 protein that are deubiquitinated by USP36. Our mass spectrum

analyses showed that lysine 146 (K146) and lysine 206 (K206) of Snail1 were two amino acid residues deubiquitinated by USP36 (Fig. 3g and Supplementary Fig. S2a). Further validation experiments showed that ectopic expression of USP36 could effectively remove polyubiquitination of Snail1^WT, but not Snail1^K146R/K206R (Snail1^2KR) (Fig. 3h). As expected, ectopic expression of USP36 could partially remove polyubiquitination of Snail1^K146R or Snail1^K206R (Fig. 3h). In addition, ectopic expression of USP36 led to upregulated expression of Snail1^WT,

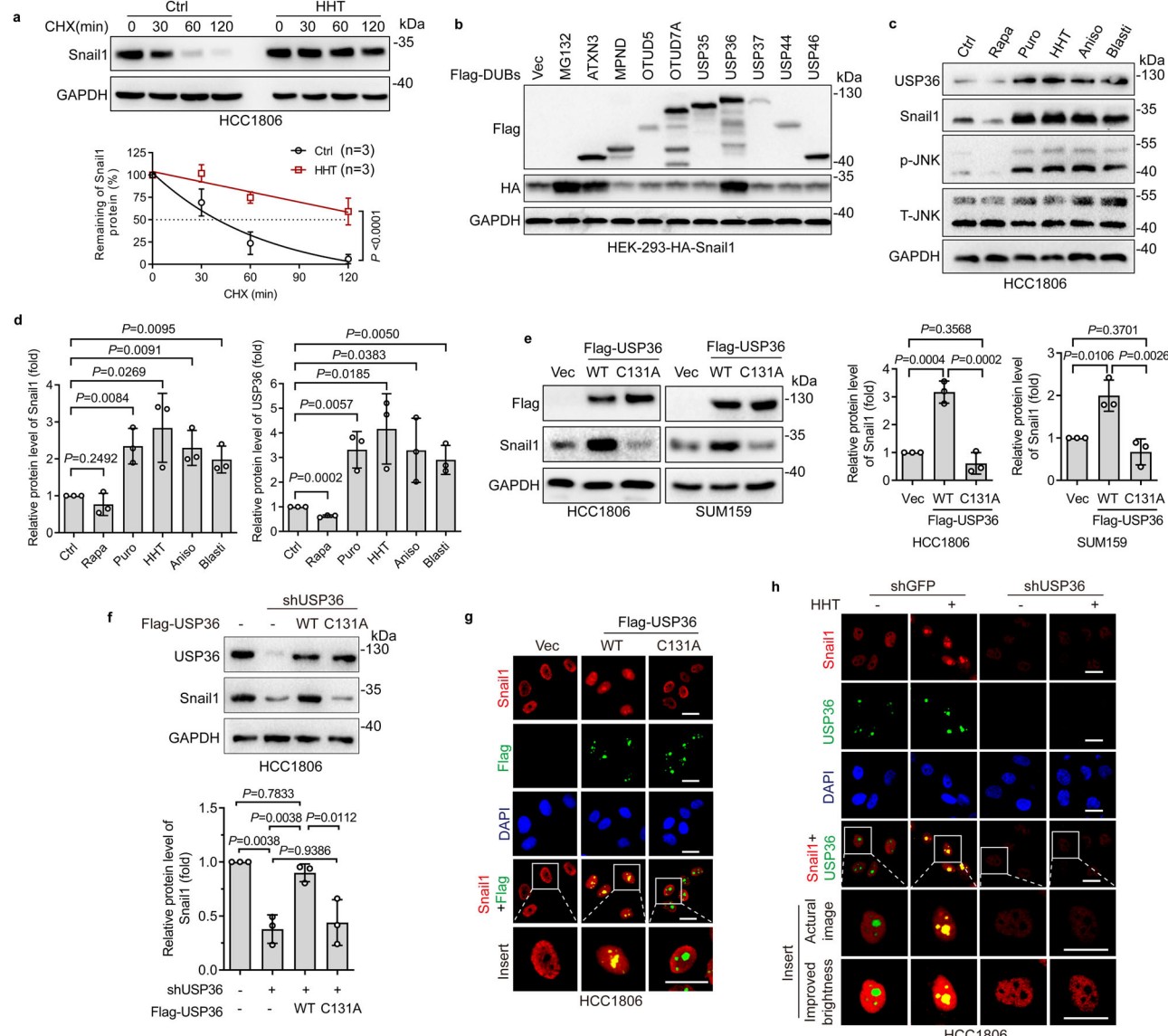

**Fig. 2 | Ribotoxic stress upregulates USP36 to promote nucleolar Snail1 accumulation. a** HCC1806 cells were treated with or without HHT (20 ng/mL and hereafter) for 24 h prior to cycloheximide (CHX, 50 μg/mL) treatment for an indicated time interval. Cells were subjected to western blot analyses. The plots of Snail1 protein half-life were presented (*n* = 3 biologically independent samples). **b** HEK-293 cells expressing HA-Snail1 were transfected with an indicated deubiquitinase or vector control (Vec) for 48 h. Cells were subjected to western blot analyses. MG132-treated cell lysates were used as a control. **c**, **d** HCC1806 cells were treated with rapamycin (Rapa, 20 nM), puromycin (Puro, 200 ng/mL), HHT, anisomycin (Aniso, 50 ng/mL), or blasticidin (Blasti, 2 μg/mL) for 24 h. Cells were subjected to western blot analyses (**c**). The Snail1 or USP36 protein levels were quantified (**d**, *n* = 3 biologically independent samples). T-JNK, total JNK; p-JNK, phospho-JNK (Thr183/Tyr185). **e** HCC1806 or SUM159 cells were infected with a recombinant lentivirus carrying wild-type (WT) USP36 or USP36-C131A mutant for

48 h. Cells were subjected to western blot analyses. The Snail1 protein levels were quantified (*n* = 3 biologically independent samples). **f** HCC1806 expressing shUSP36 were infected with a recombinant lentivirus carrying WT USP36 or USP36-C131A mutant for 48 h. Cells were subjected to western blot analyses. The Snail1 protein levels were quantified (*n* = 3 biologically independent samples). **g** HCC1806 cells were infected with a recombinant lentivirus carrying Flag-USP36 (WT or C131A) for 48 h. Cells were subjected to immunofluorescence staining for endogenous nucleolar Snail1 co-localized with Flag-USP36. **h** HCC1806 cells expressing shUSP36 or shGFP were treated with or without HHT for 24 h. Cells were subjected to immunofluorescence staining analyses. These experiments have been repeated for three times with similar results (**b**, g, **h**). Data were presented as mean ± SD and comparisons were performed with one-way ANOVA with Tukey's test (**e**, **f**), two-way ANOVA with Bonferroni's test (**a**), and unpaired two-tailed Student's *t* test (**d**). Scale bar, 25 μm.

Snail1$^{K146R}$, or Snail1$^{K206R}$, but not Snail1$^{2KR}$ (Fig. 3i, j). Notably, while Snail1$^{2KR}$ protein was more stable than Snail1$^{WT}$, ectopic expression of USP36 significantly upregulated Snail1$^{WT}$ protein half-life, but not Snail1$^{2KR}$ (Fig. 3k). Consistent with the effects of USP36 on Snail1 protein expression, HHT also markedly upregulated the expression of Snail1$^{WT}$, but not Snail1$^{2KR}$ (Fig. 3l, m).

Taken together, these results indicate that USP36 is a bona fide nucleolar deubiquitinase of Snail1 and that K146/K206 are two key amino acid residues deubiquitinated by USP36. Ribotoxic stress

elevates USP36 expression which in turn deubiquitinates and stabilizes Snail1 in the nucleolus.

**Lys157 of Snail1 is essential for Snail1-USP36 complex formation and ribotoxic stress-induced Snail1 nucleolar accumulation**
Our abovementioned data indicated that K146/K206 on Snail1 are two key amino acid residues deubiquitinated by USP36. However, immunoprecipitation analyses showed that Snail1$^{WT}$ and Snail1$^{2KR}$ had a similar binding ability with USP36 (Supplementary Fig. S2b),

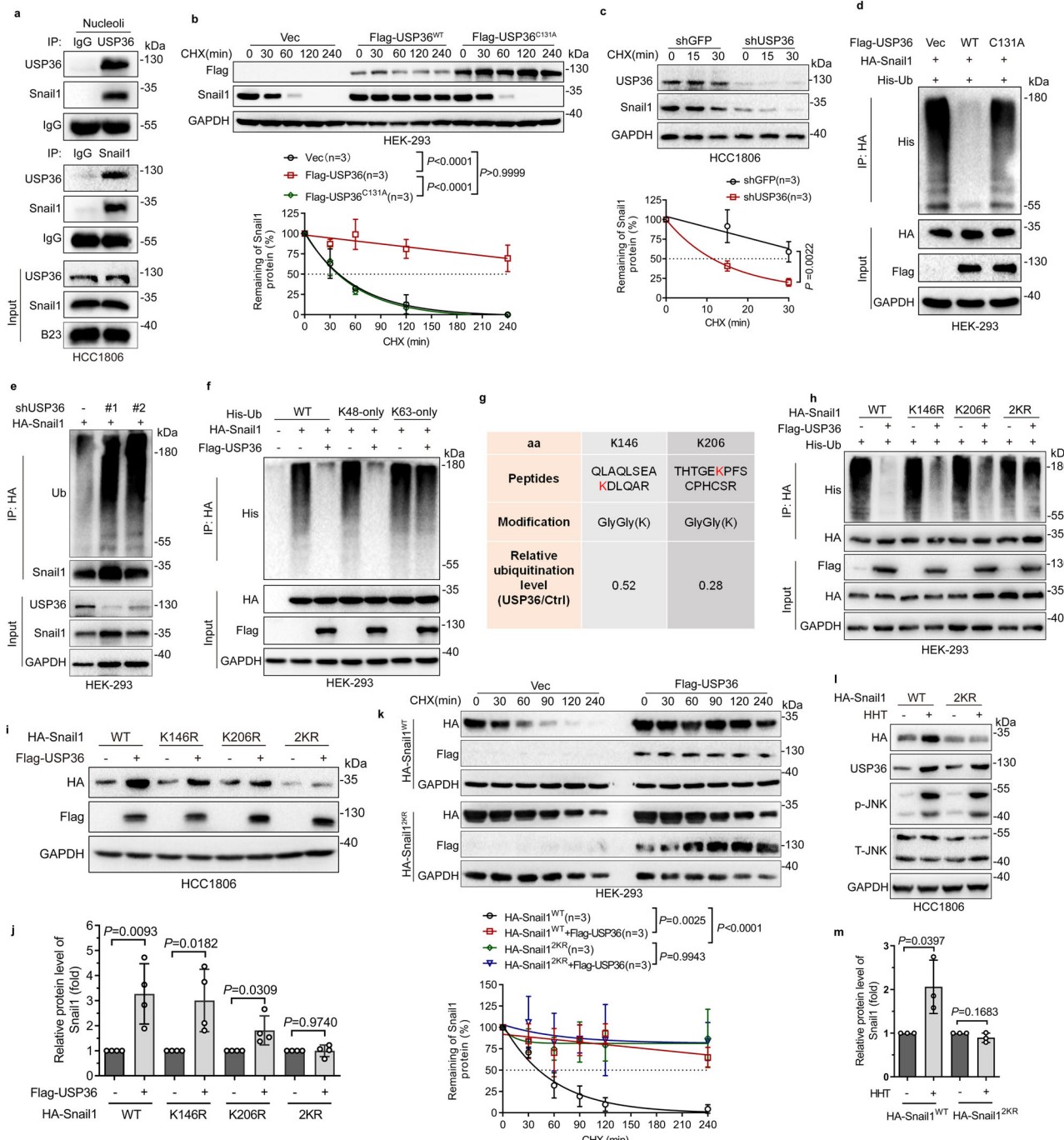

**Fig. 3 | USP36 deubiquitinates Snail1 on Lys146 and Lys206 to stabilize nucleolar Snail1. a** Cell lysates from purified nucleoli of HCC1806 cells were subjected to immunoprecipitation-western blot analyses. **b, c** HEK-293 cells expressing Flag-USP36 (WT or C131A) (**b**) or expressing shUSP36 or shGFP (**c**) were treated with cycloheximide (CHX, 50 μg/mL and hereafter) for an indicated time interval. Cells were subjected to western blot analyses. The plots of Snail1 protein half-life were presented (*n* = 3 biologically independent samples). **d–f** HEK-293-HA-Snail1 cells were transfected with Flag-USP36 (WT or C131A) in the presence of His-Ub for 48 h in (**d**) or HEK-293 cells expressing shUSP36 (#1 or #2) or shGFP were transfected with HA-Snail1 for 48 h (**e**) or HEK-293-HA-Snail1 cells were transfected with Flag-USP36 in the presence of His-Ub (WT, K48-only, or K63-only) for 48 h (**f**). Cells were treated with proteasome inhibitor MG132 (10 μM and hereafter) for 6 h followed by immunoprecipitation-western blot analyses. **g** Two amino acid residues (K146 and K206) of Snail1 deubiquitinated by USP36 were shown. **h** HEK-293 cells express indicated plasmids for 48 h. Cells were treated with MG132 for 6 h followed by immunoprecipitation-western blot analyses. **i, j** HCC1806 cells express HA-Snail1 (WT, K146R, K206R, or 2KR) and Flag-USP36 for 48 h. Cells were subjected to western blot analyses (i). The Snail1 protein levels were quantified (**j**, *n* = 4 biologically independent samples). **k** HEK-293 cells expressing HA-Snail1-WT or HA-Snail1-2KR were transfected with Flag-USP36 or Vec for 48 h. Cells were treated with CHX for an indicated time interval followed by western blot analyses. The plots of Snail1 protein half-life were presented (*n* = 3 biologically independent samples). **l, m** HCC1806 cells expressing HA-Snail1-WT or HA-Snail1-2KR were treated with or without HHT for 24 h. Cells were subjected to western blot analyses (**l**). The Snail1 protein levels were quantified (**m**, *n* = 3 biologically independent samples). These experiments have been repeated for three times with similar results (**a, d–f, h**). Data were presented as mean ± SD and comparisons were performed with two-way ANOVA with Tukey's (**b, k**) or Bonferroni's (**c**) test and unpaired two-tailed Student's *t* test (**j, m**).

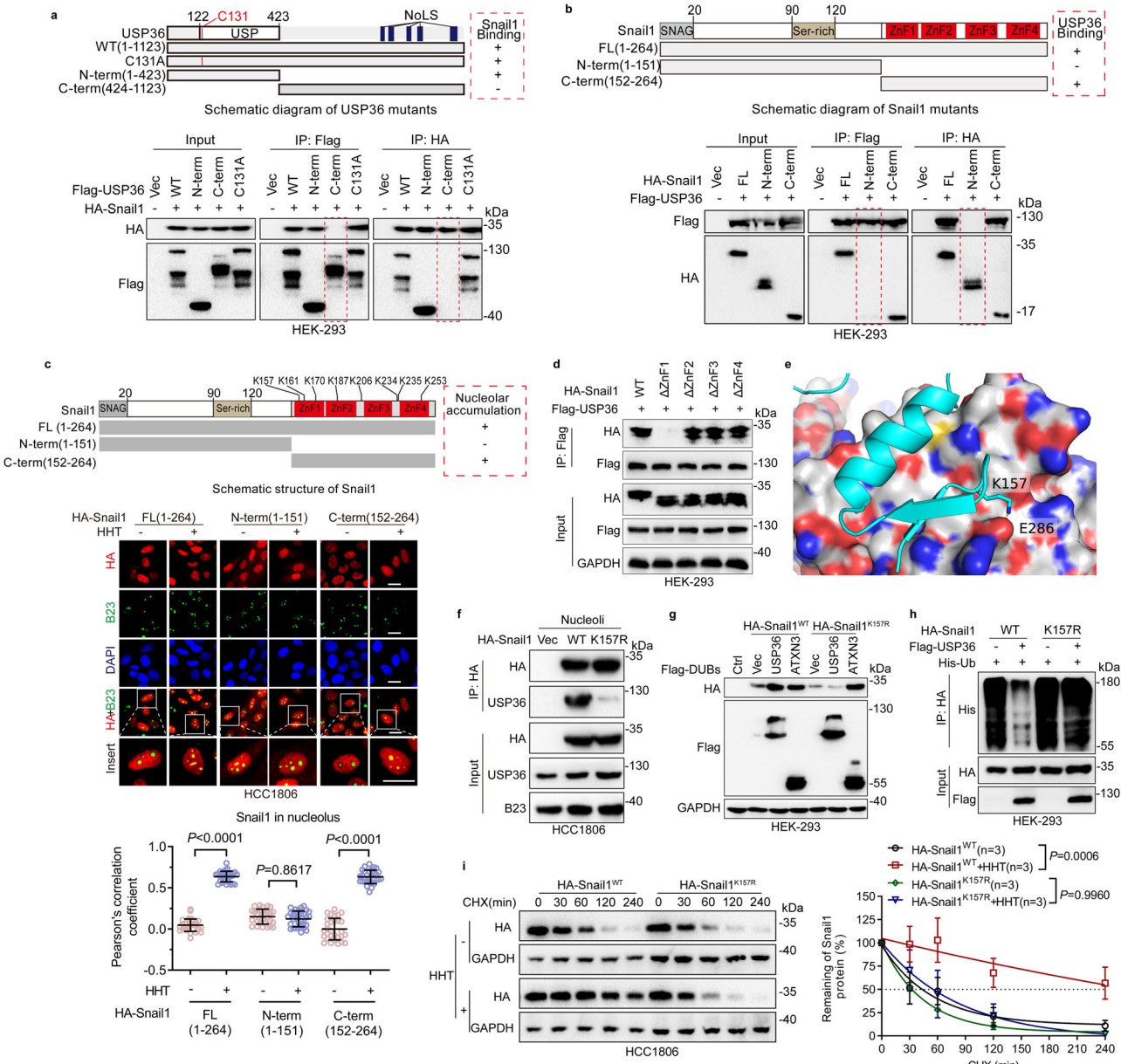

**Fig. 4 | Lys157 of Snail1 is essential for stable USP36-Snail1 protein complex formation. a, b** HEK-293 cells were co-transfected with indicated plasmids for 24 h. Cells were treated with proteasome inhibitor MG132 (10 μM and hereafter) for 6 h followed by immunoprecipitation-western blot analyses. **c** HCC1806 cells expressing an indicated Snail1 mutant were treated with or without HHT (20 ng/mL and hereafter) for 24 h followed by immunofluorescence staining. Quantification of the co-localization between Snail1 and B23 using Pearson's correlation coefficient. Quantification was carried out on 30 cells derived from three independent experiments. **d** HEK-293 cells were co-transfected with indicated plasmids for 48 h. Cells were treated with MG132 for 6 h followed by immunoprecipitation-western blot analyses. **e** The projected structure of the USP domain comprising of linear amino acid sequences of USP36 (122–423 aa) was obtained from the Alphafold2 analyses[56], which was then used in protein-protein docking (ZDOCK[30]) with the Snail1 protein crystal structure (3W5K [https://doi.org/10.2210/pdb3W5K/pdb]). ZDOCK predicted that the lysine 157 (K157) of Snail1 protein is critical for its

interaction with USP36. **f** HCC1806 cells were transfected with HA-Snail1$^{WT}$ or HA-Snail1$^{K157R}$ for 48 h. Cells were treated with MG132 for 6 h followed by immunoprecipitation-western blot analyses. **g** HEK-293 cells expressing HA-Snail1$^{WT}$ or HA-Snail1$^{K157R}$ were transfected with Flag-USP36 or Flag-ATXN3 for 48 h. Cells were subjected to western blot analyses. **h** HEK-293 cells expressing His-Ub and HA-Snail1$^{WT}$ or HA-Snail1$^{K157R}$ were transfected with or without Flag-USP36 for 48 h. Cells were treated with MG132 for 6 h followed by immunoprecipitation-western blot analyses. **i** HCC1806 cells expressing HA-Snail1$^{WT}$ or HA-Snail1$^{K157R}$ were treated with or without HHT for 24 h prior to cycloheximide (CHX, 50 μg/mL) treatment for an indicated time interval. Cell lysates were subjected to western blot analyses and plots for protein half-life were presented (*n* = 3 biologically independent samples). These experiments have been repeated for three times with similar results (**a, b, d–h**). Data were presented as mean ± SD and comparisons were performed with two-way ANOVA with Tukey's (**i**) and unpaired two-tailed Student's *t* test) (**c**). Scale bar, 25 μm.

suggesting that K146 and K206 are not required for USP36 binding. To address whether USP36 binds to ubiquitinated Snail1, we employed a specific E1 inhibitor, TAK-243, to block protein ubiquitination. As shown in Supplementary Fig. S2c, USP36 effectively interacted with Snail1 regardless of the status of its ubiquitination. Notably, USP36$^{C131A}$, defective in deubiquitinase activity, was fully capable of interaction

with Snail1 (Supplementary Fig. S2c). To identify binding segments important for USP36-Snail1 interaction, we constructed a series of truncation mutants. As shown in Fig. 4a, b, the N-terminus of USP36 (1–423 aa) interacted with the C-terminus of Snail1 (152–264 aa). In addition, immunofluorescence staining data showed that HHT could induce the nucleolar accumulation of the HA-Snail1 (152–264 aa), but

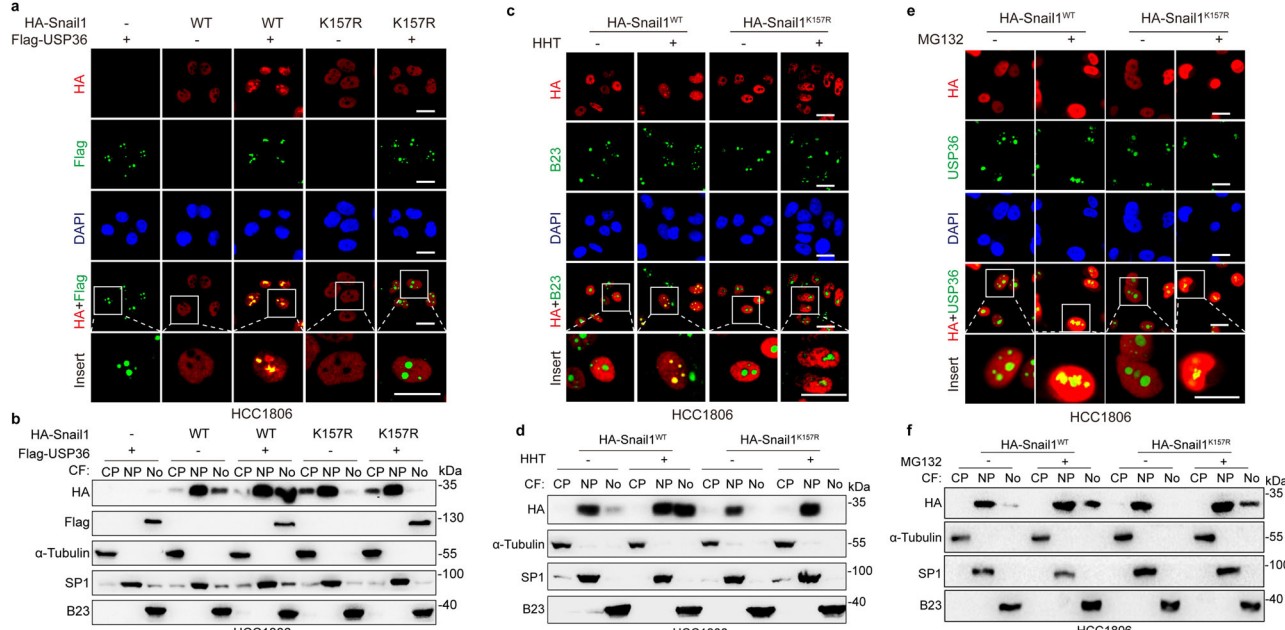

**Fig. 5 | Lys157 of Snail1 is essential for Snail1 nucleolar accumulation.**
**a**, **b** HCC1806 cells expressing HA-Snail1$^{WT}$ or HA-Snail1$^{K157R}$ were infected with a recombinant lentivirus carrying Flag-USP36 for 48 h, followed by immuno-fluorescence staining analyses (**a**) or a procedure of cellular fractionation, which were then subjected to western blot analyses for expression of Snail1 in CP Cytoplasm, NP Nucleoplasm, No Nucleolus (**b**). **c**, **d** HCC1806 cells stably expressing HA-Snail1$^{WT}$ or HA-Snail1$^{K157R}$ were treated with or without HHT (20 ng/mL) for 24 h. Cells were subjected to immunofluorescence staining (**c**) or a procedure of cellular fractionation, which were then subjected to western blot analyses for expression of Snail1 in CP, NP, or No (**d**). **e**, **f** HCC1806 cells expressing HA-Snail1$^{WT}$ or HA-Snail1$^{K157R}$ were treated with or without proteasome inhibitor MG132 for 6 h. Cells were subjected to immunofluorescence staining (**e**) or a procedure of cellular fractionation, which were subjected to western blot analyses for expression of Snail1 in CP, NP, or No (**f**). These experiments have been repeated for three times with similar results (**a**–**f**). Scale bar, 25 μm.

not HA-Snail1 (1–151 aa) (Fig. 4c), indicating that the Snail1 C-terminus consisting of the Zinc-Finger domains is responsible for HHT-induced Snail1 nucleolar accumulation. Indeed, further experiments showed that the C2H2-type 1 Zinc finger (154–176 aa, ZnF1) of Snail1 protein is responsible for USP36-Snail1 interaction (Fig. 4d).

To further identify key amino acid residue(s) in the ZnF1 of Snail1 for stable protein complex formation with USP36, we employed ZDOCK protein-protein docking analyses[30], which predicted that the K157 in the ZnF1 of Snail1 protein is most likely involved in its inter-action with USP36 (Fig. 4e). We then examined the impact of Snail1$^{K157R}$ on its ability to bind USP36. As shown in Fig. 4f, USP36 readily formed stable protein complexes with Snail1$^{WT}$, but not Snail1$^{K157R}$, in the nucleolus. Consistently, USP36 upregulated expression of Snail1$^{WT}$, but not Snail1$^{K157R}$ (Fig. 4g). By contrast, ectopic expression of ATXN3 upregulated both Snail1$^{WT}$ and Snail1$^{K157R}$ (Fig. 4g), suggesting that K157 is critical for Snail1-USP36 interaction. Furthermore, ectopic expression of USP36 led to the robust deubiquitylation of Snail1$^{WT}$, but not Snail1$^{K157R}$ (Fig. 4h). Notably, while Snail1$^{WT}$ and Snail1$^{K157R}$ exhibited a similar half-life in the absence of ribotoxic stress, HHT significantly prolonged the protein half-life of Snail1$^{WT}$, but not Snail1$^{K157R}$ (Fig. 4i), indicating that USP36–Snail1 interaction is critically important in the stabilization of Snail1 in the nucleoli upon ribotoxic stress.

Next, we investigated whether USP36-Snail1 interaction is required for Snail1 nucleolar accumulation. As shown in Fig. 5a, c, either ectopic expression of USP36 or HHT treatment led to the nucleolar accumulation of Snail1$^{WT}$, but not Snail1$^{K157R}$, which was confirmed by the cellular fractionation assays (Fig. 5b, d). Consistent with this observation, only K to R mutation on K157, among the eight lysine residues in the Snail1 C-terminus Zinc-Finger domains (152–264) (Fig. 4c), was unable to be induced to nucleolar accumulation by HHT (Fig. 5c and Supplementary Fig. S3).

We then explored whether USP36 is involved in Snail1 transloca-tion from the nucleoplasm to the nucleolus. We first established an

H1299 stable cell line expressing Snail1-GFP fusion protein. As shown in Supplementary Fig. S4a–c, while Snail1-GFP fusion protein was pri-marily localized in the nucleoplasm, either ectopic expression of USP36 or proteasome inhibitor MG132 treatment robustly promoted nucleolar Snail1 accumulation. Photobleaching analyses, which were often used to address nucleolar translocation, such as c-Myc, p21, and H2B nucleolar entry[31–33], showed that nucleoplasmic Snail1-GFP was diffused to the nucleolus after photobleaching (Supplementary Fig. S4b, c). Notably, the Snail1$^{K157R}$ mutant, which is unable to bind to USP36, was also accumulated in the nucleolus upon MG132 treatment (Fig. 5e, f), suggesting that although USP36 is dispensable for Snail1 nucleolar entry, it is essential for Snail1 protein stabilization and accumulation in the nucleolus.

Together, these results indicate that K157 of Snail1 is essential for Snail1-USP36 protein complex formation and Snail1 nucleolar accu-mulation upon ribotoxic stress.

## Ribotoxic stress promotes USP36 expression via activation of the JNK-HSF1 signaling

We then explored the molecular mechanism with which ribotoxic stress promotes USP36 expression. It is well-known that ribotoxic stress can trigger the JNK/p38 pathway to regulate gene transcription[24,34]. We thus speculated that ribotoxic stress promotes USP36 expression via activating the JNK/p38 signaling. Indeed, HHT and several ribotoxic stress inducers, including anisomycin, pur-omycin, and blasticidin, activated JNK signaling in the upregulation of USP36 and Snail1 protein expression (Fig. 2c, d). By contrast, rapa-mycin, an mTOR inhibitor to inhibit protein synthesis, failed to either activate JNK or to upregulate USP36 and Snail1 protein expression (Fig. 2c, d), in keeping with our previous observation that rapamycin failed to promote Snail1 nucleolar accumulation (Fig. 1b, c). Moreover, we found that ribotoxic stress induced by HHT can significantly increase USP36 steady-state mRNA levels (Supplementary Fig. S5a).

Importantly, while a selective p38 inhibitor, SB203580, failed to block the HHT-induced upregulation of USP36 and Snail1, inhibition of JNK by SP600125 completely inhibited HHT-induced upregulation of the mRNA and protein expression of USP36 (Supplementary Fig. S5b–d). Furthermore, JNK inactivation completely inhibited HHT-induced Snail1 nucleolar accumulation (Supplementary Fig. S5e). Together, these results indicate that activation of JNK is responsible for ribotoxic stress-induced USP36 expression and nucleolar Snail1 accumulation.

Next, we investigated the molecular mechanism by which JNK regulates USP36 expression. It has been reported that JNK can regulate a subset of downstream transcription factors, including c-JUN, p53, YAP1, and HSF1 (heat shock factor 1), to impact gene transcription[35–37]. We analyzed clinical relevance and found a positive correlation between HSF1 and USP36 with regard to mRNA expression (Supplementary Fig. S6a). By PROMO database analyses[38,39], we found that there are two putative HSF1-binding elements on the *USP36* gene promoter (P1: −1880 to −1753; P2: −393 to −279) (Supplementary Fig. S6b). Chromatin immunoprecipitation (ChIP) analyses showed that HSF1 directly bound on the *USP36* gene promoter P2 element (−393 to −279) (Supplementary Fig. S6c, d). Notably, ectopic expression of JNK1 upregulated HSF1 expression as well as USP36 and Snail1 expression (Supplementary Fig. S6e). Inhibition of JNK signaling by SP600125 completely suppressed HHT-induced upregulation of HSF1, USP36, or Snail1 protein expression (Supplementary Fig. S6f). In addition, the pharmacological inhibition of HSF1 by KRIBB11 dramatically inhibited USP36 and Snail1 expression (Supplementary Fig. S6g). Importantly, KRIBB11 also markedly inhibited HHT-mediated upregulation of USP36 and Snail1 expression (Supplementary Fig. S6h).

Taken together, these results indicate that ribotoxic stress activates JNK signaling to upregulate HSF1 expression, which in turn transactivates USP36 expression.

### Activation of nucleolar USP36-Snail1 axis promotes ribosome biogenesis to promote cancer cell survival in response to ribotoxic stress

We next investigated the biological significance of the nucleolar USP36-Snail1 axis. We first examined the effects of Snail1[WT] or Snail1[K157R] on the nucleoplasmic transcriptional regulation of E-cadherin, a critical downstream target in EMT. As shown in Fig. 6a, b, ectopic expression of either Snail1[WT] or Snail1[K157R] inhibited E-cadherin mRNA and protein expression, indicating an intact EMT-regulatory function in the nucleoplasm. Importantly, Snail1[WT], capable of nucleolar localization, significantly upregulated the expression of 47S pre-rRNA (Fig. 6c, d). By contrast, Snail1[K157R], unable to be accumulated in the nucleolus, failed to do so (Fig. 6c, d). Conversely, the knockdown of Snail1 led to a marked reduction of 47S pre-rRNA expression (Fig.6e and Supplementary Fig. S7a). Notably, the knockdown of USP36 significantly inhibited the 47S pre-rRNA expression (Fig. 6f), which could be completely rescued by ectopic expression of Snail1[WT], but not Snail1[K157R] (Fig. 6g–i), indicating that nucleolar Snail1 is responsible for the knockdown of USP36-mediated downregulation of 47S pre-rRNA expression. Furthermore, HHT significantly promoted 47S pre-rRNA expression, which was effectively reversed by the knockdown of Snail1 (Fig. 6j, k and Supplementary Fig. S7b), suggesting that the nucleolar Snail1 plays a critical role in the regulation of ribosome biogenesis in the cellular response to ribotoxic stress.

Since HHT promotes 47S pre-rRNA expression, we next examined the role of the nucleolar USP36-Snail1–47S pre-rRNA axis in the regulation of HHT-induced cancer cell apoptosis. As shown in Fig. 6l and Supplementary Fig. S7c, HHT alone had a marginal effect in inducing cell apoptosis, while CX-5461, an inhibitor of RNA polymerase Pol I known to inhibit 47S pre-rRNA expression and induce cellular senescence[40,41], also had little effect on cell apoptosis. However, HHT in combination with CX-5461 significantly promoted cell apoptosis

(Fig. 6l and Supplementary Fig. S7c), indicating that 47S pre-rRNA serves as a survival factor in inhibiting HHT-induced cell apoptosis. Since Snail1 can transcriptionally upregulate 47S pre-rRNA, we postulated that HHT-induced upregulation of Snail1, which in turn upregulates 47S pre-rRNA expression, can function as a survival factor to defy ribotoxic stress. Indeed, as shown in Fig. 6m, n and Supplementary Fig. S7d, e, HTT treatment could induce little apoptosis accompanied by increased snail1 expression. Notably, the knockdown of Snail1 led to a dramatic increase in apoptosis upon HHT treatment, as evidenced by increased cleaved caspase-3 (CC3) and apoptotic cell population, both of which were completely rescued by restoration of Snail1[WT], but not Snail1[K157R] (Fig. 6m, n and Supplementary Fig. S7d, e). Similarly, the knockdown of USP36 markedly sensitized cancer cells to HHT-induced cell death, which were effectively rescued by ectopic expression of Snail1[WT] but not Snail1[K157R] (Fig. 6o, p and Supplementary Fig. S7f, g). In keeping with our finding that JNK signaling upregulates USP36 expression, a combination of HHT with SP600125, a selective inhibitor of JNK, synergistically induced apoptosis, which could be largely rescued by ectopic expression of USP36 (Fig. 6q and Supplementary Fig. S7h). Importantly, HHT in combination with Snail1 knockdown led to a robust inhibition of xenograft tumor growth, which could be completely rescued by ectopic expression of Snail1[WT] but not by Snail1[K157R] (Fig. 6r–t).

### Inhibition of the JNK-USP36-Snail1 signaling sensitizes solid tumor cells to HHT

HHT has widely been used in clinical treatment for leukemia with great benefits for leukemia patient outcomes[5]. However, various clinical trials have shown poor efficacy of HHT anticancer activity for solid tumors[8], the reasons for which remain unknown. Our abovementioned data indicate that activation of the nucleolar USP36-Snail1 axis promotes ribosome biogenesis to sustain cancer cell survival in response to ribotoxic stress, exemplified by HHT. We thus speculated that the JNK-USP36-Snail1 axis is a critical cellular surveillance mechanism against ribotoxic stress. To investigate this hypothesis, we examined the effects of HHT on JNK-USP36-Snail1 signaling in solid tumor cells compared to leukemia cells. As shown in Fig. 7a, b and Supplementary Fig. S8a, b, again, HHT effectively activated JNK, leading to upregulated expression of nucleolar USP36 and Snail1, accompanied by little apoptosis in triple-negative breast cancer SUM159, HCC1806, or Hs 578T cells. In sharp contrast, HTT failed to activate the JNK-USP36-Snail1 pathway, resulting in robust apoptosis in leukemia K-562, OCI-AML2, or MOLM-13 cells (Fig. 7a, b and Supplementary Fig. S8a, b). Importantly, inhibition of JNK by SP600125 significantly promoted HHT-induced cleaved caspase-3 (CC3) expression and cell apoptosis in SUM159, HCC1806, or Hs 578T cells (Fig. 7c, d and Supplementary Fig. S8c, d). SP600125 could also markedly promote anisomycin- or blasticidin-induced apoptosis in HCC1806 cells (Fig. 7e, f and Supplementary Fig. S8e, f), suggesting that the JNK-USP36-Snail1 axis functions as a surveillance mechanism, not only in response to HHT but also to other ribotoxic stress inducers. Importantly, a combination of HHT and SP600125 significantly inhibited HCC1806 cell-derived xenograft tumor growth in vivo (Fig. 7g, h). Further analyses showed that SP600125 also markedly inhibited HHT-mediated upregulation of USP36 and Snail1 expression and Snail1 nucleolar localization in vivo (Fig. 7i–k and Supplementary Fig. S8g).

Together, these results indicate that activation of the JNK-USP36-Snail1 axis leads to solid tumor cell resistance to HHT, and targeting JNK is a potential strategy to overcome solid tumor resistance to HHT.

### Discussion

Tumor growth requires rapid protein synthesis in the ribosomes. Since ribosome function is highly elevated in tumor cells to meet the need for tumor growth, inhibition of the ribosome function has been considered an effective strategy for cancer therapy. Currently, several

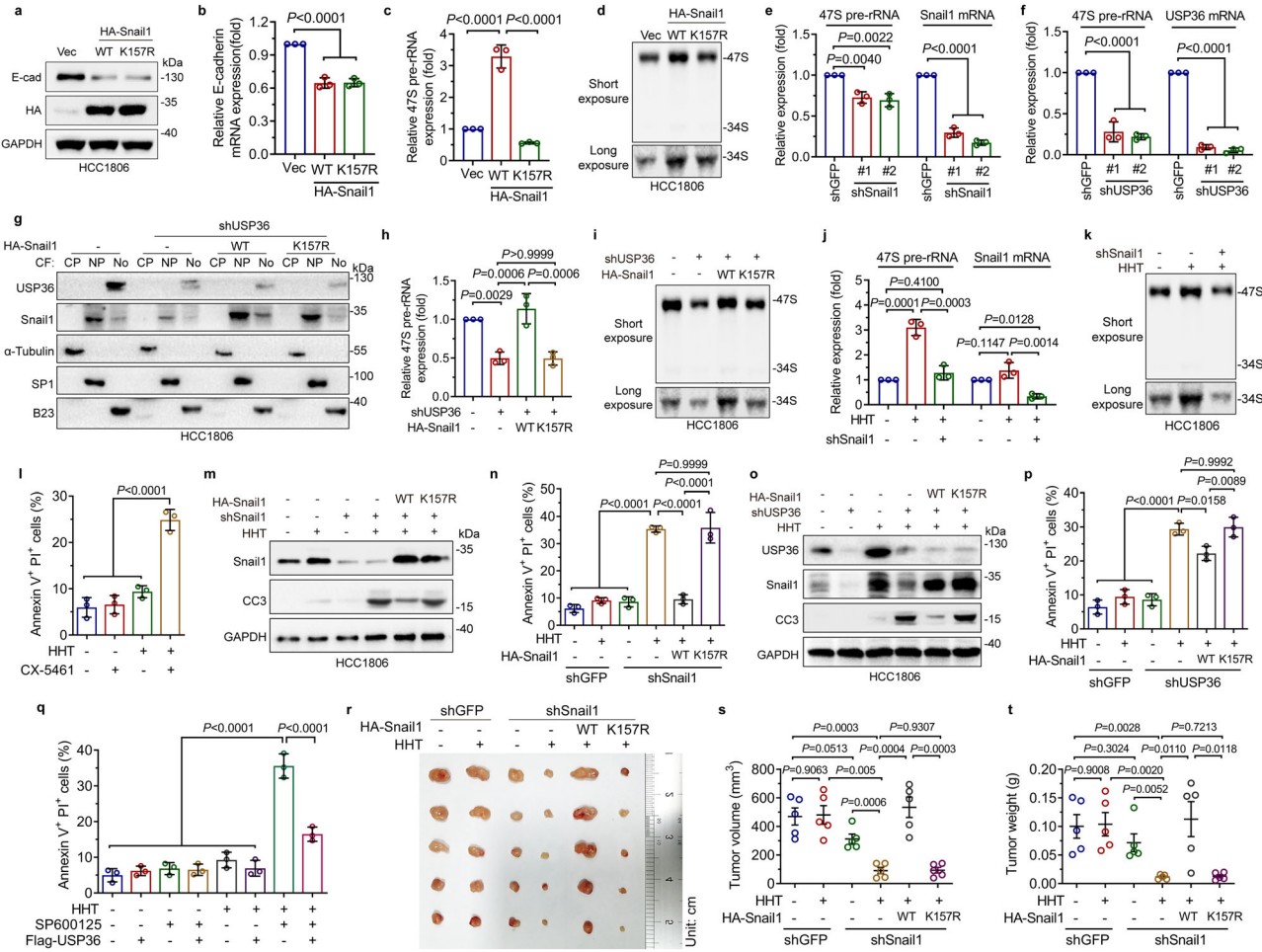

**Fig. 6 | Activation of the nucleolar USP36-Snail1 axis promotes ribosome bio-genesis to promote cancer cell survival upon ribotoxic stress. a–d** HCC1806 cells expressing HA-Snail1^WT or HA-Snail1^K157R were subjected to western blot (**a**), qPCR (**b**, **c**, *n* = 3 biologically independent samples), or northern blot (**d**) analyses. **e**, **f** HCC1806 cells expressing shSnail1 (#1 or #2) or shUSP36 (#1 or #2) were subjected to qPCR analyses (*n* = 3 biologically independent samples). **g–i** HCC1806 cells expressing indicated plasmids were subjected to cellular fractionation (**g**), qPCR (**h**, *n* = 3 biologically independent samples), or northern blot analyses (**i**). CP Cytoplasm, NP Nucleoplasm; No Nucleolus. **j**, **k** HCC1806 cells expressing shSnail1 were treated with or without HHT (20 ng/mL and hereafter) for 24 h. Cells were subjected to qPCR (**j**, *n* = 3 biologically independent samples) or northern blot analyses (**k**). **l** HCC1806 cells were treated with or without HHT in the presence or absence of CX-5461 (200 nM) for 48 h, followed by PI-Annexin V staining analyses (*n* = 3 biologically independent samples). **m–p** HCC1806 cells expressing indicated

plasmids were treated with or without HHT followed by western blot analyses (**m**, **o**) or FACS analyses (**n**, **p**, *n* = 3 biologically independent samples). CC3: Cleaved-Caspase-3. **q** HCC1806 cells expressing Flag-USP36 were treated with or without HHT in the presence or absence of a JNK inhibitor SP600125 (20 μM) for 48 h. Cells were subjected to FACS analyses (*n* = 3 biologically independent samples). **r–t** HCC1806 cells (5 × 10⁵), as indicated, were subcutaneously inoculated in 5-week-old female BALB/c nude mice (*n* = 5/group). On day 3 after inoculation, mice were intraperitoneally (i.p) injected with HHT (1 mg/kg) daily. Mice were monitored for tumor size and sacrificed on day 14 after i.p. Dissected tumors were photographed (**r**). Tumor volume (**s**) and weight (**t**) were presented. These experiments have been repeated for three times with similar results (**a**, **d**, **g**, **i**, **k**, **m**, **o**). Data were presented as mean ± SD (**b**, **c**, **e**, **f**, **h**, **j**, **l**, **n**, **p**, **q**) or SEM (**s**, **t**). Comparisons were performed with one-way ANOVA with Tukey's test (**b**, **c**, **e**, **f**, **h**, **j**, **l**, **n**, **p**, **q**) and unpaired two-tailed Student's *t* test (**s**, **t**).

ribosome inhibitors, such as anisomycin or lactimidomycin, have been shown to effectively inhibit can cell proliferation in vitro and tumor growth in vivo[42–44]. Interestingly, homoharringtonine (HHT), an effective ribosome inhibitor in blocking protein translation elongation, has been widely used for the treatment of leukemia[5]. However, HHT is ineffective in the inhibition of solid tumors, the reasons for which are unknown[8]. It is plausible that solid tumors may exist surveillance mechanisms to protect cells from ribotoxic stress.

Snail1 is a key transcription factor in the regulation of EMT, cancer stemness, drug resistance, and cell survival[21]. It has been shown that the Snail1 protein can be shuttled between the cytoplasm and nucleoplasm. GSK-3β-mediated phosphorylation of Snail1 promotes Snail1 protein cytoplasmic localization and degradation[45], whereas p21-activated kinase 1 (PAK1) promotes Snail1 phosphorylation at Ser246 and nuclear localization[46]. In this study, we show that Snail1 can be induced to localize in the nucleolus upon ribotoxic stress, which

serves as a cellular surveillance factor to promote cancer cell survival. At the molecular level, ribotoxic stress promotes USP36-dependent Snail1 stabilization in the nucleolus to facilitate ribosome biogenesis (Fig. 7l), consistent with a recent report implicating Snail1 in rRNA biosynthesis during EMT[47]. Thus, Snail1 possesses a biological function in the nucleolus independent of EMT-promoting activity.

Snail1 is a highly unstable transcriptional factor. Nuclear Snail1 protein stability is tightly controlled by several ubiquitin E3 ligases and deubiquitinases[26]. It has been shown that FBXW7 is an E3 ligase of nucleoplasmic Snail1[48]. In this study, we show that USP36 is the bona fide deubiquitinase for nucleolar Snail1. We demonstrate that while Lys157 of Snail1 is critical for USP36-Snail1 protein complex formation, Lys146 and Lys206 on Snail1 are two key amino acid residues deubiquitinated by USP36. Interestingly, USP36 has been reported to interact with the nucleolar FBXW7γ but not the nucleoplasmic FBXW7α[27]. It would be interesting to know whether FBXW7γ can

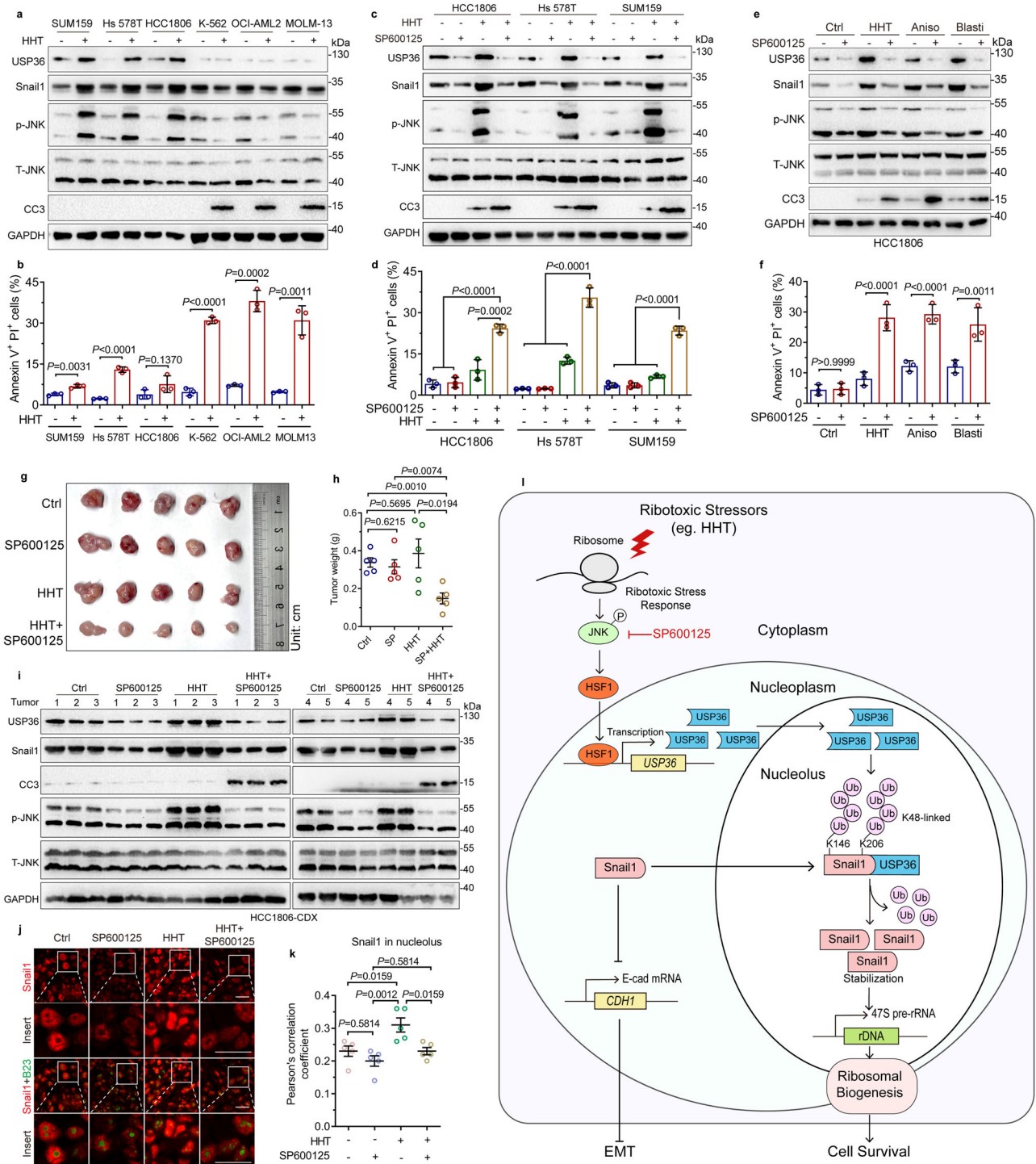

function as an E3 ligase of nucleolar Snail1 and whether there is a balance between USP36 and FBXW7γ that play a role in maintaining nucleolar Snail1 protein homeostasis.

Snail1 protein lacks a nucleolar localization signal (NoLS). Then a key question is how the Snail1 protein enters the nucleolus. Since the USP36 protein bears several NoLS[29,49], it is plausible that USP36 binds to and shuttles Snail1 into the nucleolus and stabilizes Snail1. However, our photobleaching analyses, which have been often used to address the protein nucleolar translocation, including c-Myc, p21, and H2B[31–33], show that nucleoplasmic Snail1 can diffuse to the nucleolus after photobleaching. Notably, the Snail1[K157R] mutant protein, which is unable to bind USP36, can be accumulated in the nucleolus only after

inhibition of proteasome by MG132, suggesting that although USP36 is dispensable for Snail1 nucleolar entry, it is essential for Snail1 protein stabilization and accumulation in the nucleolus.

One highlight of this study is that USP36-mediated stabilization of nucleolar Snail1 is essential for ribosome biogenesis and cancer cell survival upon ribotoxic stress. However, the knockdown of USP36-mediated cancer cell death upon ribotoxic stress can only be partly rescued by ectopic expression of wild-type Snail1, suggesting that Snail1 is important but not the only downstream effector of USP36. Notably, it has been reported that serum stimulation can upregulate USP36, leading to deubiquitination and stabilization of nucleolar c-Myc, which in turn promotes rDNA transcription and cell

**Fig. 7 | Inhibition of JNK-USP36-Snail1 Signaling sensitizes solid tumor cells to HHT. a, b** Triple-negative breast cancer (SUM159, Hs 578T, and HCC1806) and non-lymphocytic leukemia (K-562, OCI-AML2, and MOLM-13) cells were treated with or without HHT (20 ng/mL and hereafter). Cells were subjected to western blot analyses (**a**) or FACS analyses (**b**). Annexin V + /PI+ cell populations were statistically analyzed (**b**, *n* = 3 biologically independent samples). **c, d** SUM159, Hs 578T, or HCC1806 cells were treated with or without a JNK inhibitor SP600125 (20 μM and hereafter) in the presence or absence of HHT. Cells were subjected to western blot analyses (**c**) or FACS analyses (**d**). Annexin V + /PI+ cell populations were statistically analyzed (**d**, *n* = 3 biologically independent samples). **e, f** HCC1806 cells were treated with HHT, anisomycin (Aniso, 50 ng/mL), or blasticidin (Blasti, 2 μg/mL) in the presence or absence of SP600125. Cells were subjected to western blot analyses (**e**) or FACS analyses (**f**). Annexin V + /PI+ cell populations were statistically analyzed (**f**, *n* = 3 biologically independent samples). **g–k** HCC1806 cells ($5 \times 10^5$) were

subcutaneously inoculated in 5-week-old female BALB/c nude mice (*n* = 5/group). On day 3 after inoculation, mice were intraperitoneally (i.p) injected with SP600125 (15 mg/kg) and/or HHT (1 mg/kg) daily. Dissected tumors were photographed on day 17 after i.p (**g**). Tumor weights (**h**) were presented. The xenograft tumor samples were subjected to western blot analyses (**i**) or immunofluorescence staining (**j**). Pearson's correlation coefficient was used to qualify the co-localization of Snail1 and B23, using images derived from immunostained tumor samples (**k**). Scale bar, 25 μm. **l** A model depicts that USP36 stabilizes nucleolar Snail1 to promote ribosome biogenesis and cancer cell survival in response to ribotoxic stress. These experiments have been repeated for three times with similar results (**a, c, e**). Data were presented as mean ± SD (**b, d, f**) or SEM (**h, k**). Comparisons were performed with one-way ANOVA with Tukey's test (**d, f, k**) and unpaired two-tailed Student's *t* test (**b, h**). CC3 Cleaved-Caspase-3, T-JNK total JNK, p-JNK phospho-JNK (Thr183/Tyr185).

proliferation[27], raising the possibility that c-Myc might be involved in ribotoxic stress-induced ribosome biogenesis. However, our results show that while HHT- significantly upregulates nucleolar Snail1, it reduces c-Myc protein expression (Supplementary Fig. S9), in keeping with a previous report that HHT can reduce *MYC* gene transcription via suppression of NF-κB signaling[6]. Therefore, these results imply that c-Myc is unlikely to be involved in the HHT-induced upregulation of rDNA transcription and ribosome biogenesis. It is plausible that, upon ribotoxic stress, USP36 affects other factors involved in cell viability besides Snail1.

Nucleolar Snail1-induced upregulation of 47S pre-rRNA expression renders cell survival upon ribotoxic stress, the molecular mechanisms of which remain unclear. Ribotoxic stress can be caused by small molecules that bind to and impair ribosomes, ultimately leading to inflammation and apoptosis[3,50]. Notably, it has been reported that ribophagy, an intracellular autophagic process, can remove nonfunctional ribosomes to maintain cell survival upon nutrient stress[51]. Thus, it is plausible that the damaged ribosomes by ribotoxic stress inducers might be cleared by mechanisms such as ribophagy, accompanied by elevated ribosome biogenesis that would fit the need for cell survival and growth. We propose that nucleolar Snail1-mediated 47S pre-rRNA biogenesis could be critical in the maintenance of ribosome homeostasis in the cellular response to ribotoxic stress.

An important finding of this study is that activation of the JNK-USP36-nucleolar Snail1 axis serves as a general surveillance mechanism in solid tumors against ribotoxic stress. Indeed, the combination of HHT with a selective JNK inhibitor SP600125 synergistically induces apoptosis of solid tumor cells and suppresses xenograft tumor growth in vivo. By contrast, leukemia cells are sensitive to HHT due to its inability to activation of the JNK−USP36−nucleolar Snail1 axis. Therefore, our results offer a plausible explanation that why HTT exhibits little anticancer activity on solid tumors and provide a rationale for targeting the JNK−USP36−Snail1 axis in ribosome inhibition-based solid tumor treatment.

## Methods
### Ethics statement and mouse models
All animal care and animal experiments in this study were performed in accordance with China's National Legislation and the institutional ethical guidelines and were approved by the Institutional Animal Care and Use Committee of Sichuan University (IACUC). Female BALB/c nude mice (BALB/cNj-Foxn1nu/Gpt) were purchased from Gem-Pharmatech (Chengdu, China). Mice were maintained in individual cages at a room temperature of 22 ± 2 °C and humidity of 50−60%, on a 12:12 light−dark cycle (lights on at 09:00 h). Cells ($5 \times 10^5$) were subcutaneously inoculated into the right scruff of each nude mouse (*n* = 5/group). On day 3 after inoculation, mice were intraperitoneally (i.p.) injected with DMSO (5% V/V) or with SP600125 (15 mg/kg) and/or HHT (1 mg/kg) daily. Mice were monitored for tumor size daily and sacrificed on the indicated day after i.p. Tumor weight, volume, and photos

were taken. The xenograft tumor samples were subjected to western blot analyses or immunofluorescence staining analyses. Tumor size was measured with a caliper and tumor volume was calculated by $width^2 \times length \times 1/2$. The maximal tumor size permitted by the IACUC of Sichuan University is 20 mm at the largest diameter in mice. The maximal tumor size in our animal experiments did not exceed the permitted maximal tumor size.

### Cell culture and reagents
HCC1806 (CRL-2335), K-562 (CCL-243), HEK-293 (CRL-1573), and Hs 578T (HTB-126) were obtained from ATCC (Manassas, VA, USA). The SUM159 (CL-0622) and NCI-H1299 (CL-0165) were obtained from Procell Life Science&Technology (Wuhan, China). HEK-293FT (R70007) was obtained from Thermo Fisher Scientific (Waltham, MA, USA). A549 (BNCC337696), OCI-AML2 (BNCC341618), and NOLM-13 (BNCC100895) were obtained from BeNa Culture Collection (Beijing, China). All cell lines used in this study were routinely tested to be negative for mycoplasma contamination and were kept at low passages to maintain their identity and were authenticated by morphology check and growth curve analysis.

HEK-293, HEK-293FT, A549, H1299, Hs 578T, HCC1806, and SUM159 cells were cultured in DMEM medium (Gibco, Rockville, MD, USA), whereas K-562, OCI-AML2, and NOLM-13 cells were cultured in RPMI-1640 medium. All cells were grown in a medium supplemented with 10% fetal bovine serum (FBS; HyClone, Logan, UT, USA), 100 units/mL penicillin (Gibco, Rockville, MD, USA), and 100 μg/mL streptomycin (Gibco, Rockville, MD, USA). Cells were grown in a humidified 37 °C incubator in a 5% $CO_2$ atmosphere. Cells at 60–70% confluence were treated with an indicated chemical compound. Homoharringtonine (S9015), anisomycin (S7409), rapamycin (S1039), G418 (S3028), puromycin (S7417), blasticidin (S7419), SP600125 (S1460), SB203580 (S1076), MG132 (S2619), KRIBB11 (HY-100872), and CX-5461 (S2684) were purchased from Selleck Chemicals (Houston, USA). Tunicamycin (ab120296) was purchased from Abcam (Cambridge, MA, USA). Cycloheximide (CHX, C7698) was purchased from Sigma-Aldrich (St. Louis, USA).

### Plasmids transfection, lentiviral infection, and RNA interference
Cells at 70% confluence were transfected using Lipofectamine 2000 (Invitrogen, Carlsbad, CA, USA). Expression plasmids were used in this study including human Flag-ATXN3, Flag-MPND, Flag-OTUD5, Flag-OTUD7A, Flag-USP35, Flag-USP36, Flag-USP37, Flag-USP44, Flag-USP46, Flag-USP36$^{C131A}$, Flag-USP36$^{N-term(1–423)}$, Flag-USP36$^{C-term(424–1123)}$, Flag-JNK1, HA-Snail1, HA-Snail1$^{K146R}$, HA-Snail1$^{K206R}$, HA-Snail1$^{K146R/K206R}$, HA-Snail1$^{K157R}$, HA-Snail1$^{K170R}$, HA-Snail1$^{K187R}$, HA-Snail1$^{K206R}$, HA-Snail1$^{K234R}$, HA-Snail1$^{K235R}$, HA-Snail1$^{K253R}$, HA-Snail1$^{\triangle ZnF1}$, HA-Snail1$^{\triangle ZnF2}$, HA-Snail1$^{\triangle ZnF3}$, HA-Snail1$^{\triangle ZnF4}$, HA-Snail1$^{N-term(1–151)}$, HA-Snail1$^{C-term(152–264)}$, Snail1-GFP, His-Ub, His-Ub-K48-only, and His-Ub-K63-only (The K48-only or K63-only ubiquitin mutant only forms polyubiquitin chains linked through lysine 48 or lysine 63). Recombinant lentiviruses were

amplified by transfection of HEK-293FT cells with pMD2.G and psPAX2 packaging plasmids and lentiviral expression plasmid using Lipofectamine 2000. Viruses were collected at 60 h after transfection. Cells at 60% confluence in the presence of 10 µg/mL polybrene were infected with recombinant lentivirus encoding or an empty vector, followed by 12 h of incubation at 37 °C with 5% $CO_2$. Lentiviral-based shRNAs targeting Snail1, USP36, or green fluorescent protein (GFP) were constructed into a pLKO.1-puromycin lentiviral vector. (The primer sequences were listed in Supplementary Table 4).

## Cellular fractionation and Snail1 ubiquitylation assays

Cellular fractionation of cytoplasm, nucleoplasm, and nucleoli was performed as described[27,52]. Briefly, cells were collected, washed twice with cold PBS, and resuspended in 1 mL buffer A (10 mM Tris-HCl PH 7.8; 10 mM KCl; 1.5 mM $MgCl_2$; 0.5 mM DTT) for 10 min on ice; The cells were homogenized using tight pestle douncer followed by spinning down at $228 \times g$ for 5 min at 4 °C. The supernatant was a cytoplasmic fraction. The nuclear pellets were washed with buffer A and then resuspended in 1 mL buffer S1 (0.25 M sucrose; 10 mM $MgCl_2$), layered over 1 mL buffer S2 (0.35 M sucrose, 0.5 mM $MgCl_2$), and centrifuged at $1430 \times g$ for 10 min at 4 °C. Resuspend the clean pelleted nuclei in 0.5 mL buffer S2; Sonicated for $12 \times 10$ s (with 10-s rest between each sonication) at 20% full power in an ice bath to prevent overheating of the sample. Layer the sonicated sample over 0.5 mL buffer S3 (0.88 M sucrose; 0.5 mM $MgCl_2$). Spin at $3000 \times g$ for 20 min at 4 °C. Collect the supernatant (Nucleoplasmic fraction) and the pellet (the nucleolar fraction). Resuspend the pelleted nuclei in 0.5 mL buffer S1; Layer over 0.5 mL 0.35 M buffer S2; Spin at 2500 rpm ($1430 \times g$) for 10 mins at 4 °C. For Western blot analyses, The insoluble fraction containing nucleoli was lysed in 1×SDS Sample Buffer (#7722, CST) and sonicated if necessary. For the co-immunoprecipitation assay, the nucleoli were lysed in high salt RIPA buffer (50 mM Tris pH 7.5, 500 mM NaCl, 1% Nonidet P-40, 0.5% deoxycholate, and proteasome inhibitors).

For Snail1 protein ubiquitylation assays, the collected cells were lysed in a pre-boiled denaturing cell lysis buffer (50 mM pH7.4 Tris-HCl, 70 mM β-ME, and β-ME is added for fresh). Cell lysates were boiled for 10 min, and add 4 times the volume of dilution buffer (20 mM pH 7.4 Tris-HCl, 300 mM NaCl, 1 mM EDTA, 1 mM EGTA, 1% Triton X-100, 2.5 mM sodium pyrophosphate, 1 mM β-glycerophosphate, 1 mM $Na_3VO_4$, 1 µg/mL leupeptin). Cell lysates were subjected to sonication and centrifugation at $12,000 \times g$ for 30 min followed by immunoprecipitation with anti-HA agarose beads and western blot analyses.

## Western blot, co-immunoprecipitation, and immuno-fluorescence staining analyses

For western blot analyses, cells were collected, washed twice with cold PBS, and lysed in 1× SDS Sample Buffer (#7722, Cell Signaling Technology, USA) according to the manufacturer's protocol supplement with proteasome inhibitor cocktail. Equal amounts of protein were loaded, separated by SDS-PAGE, and transferred to PVDF membranes (Millipore, Darmstadt, Germany). Membranes were blocked in 4% nonfat dry milk and hybridized to a primary antibody and horseradish peroxidase (HRP)-conjugated secondary antibody for subsequent detection by chemiluminescence (Bio-Rad ChemiDoc XRS+, Bio-Rad). Gel and blot images were analyzed using Image Lab Software 5.0. Antibodies for Snail1 (#3895, 1:1000), GAPDH (#5174, 1:2000), p38 (#9212, 1:1000), p-p38 (#9216, 1:1000), JNK (#9252, 1:1000), p-JNK (#9251, 1:1000), Tubulin (#3873, 1:1000), SP1 (#5931, 1:1000), HA-Tag (#5017, 1:1000), Cleaved-caspase 3 (CC3, #9654, 1:1000) and Flag-Tag (#8146, 1:1000) were purchased from Cell Signaling Technology (Danvers, MA, USA). Antibody for USP36 (14783-1-AP, 1:1000) was purchased from Proteintech (Chicago, IL, USA). Antibody for B23 (MA5-12508, 1:1000) was purchased from ThermoFisher Scientific (Wilmington, DE, USA). Antibodies for c-Myc (CY5150, 1:1000),

HSF1(CY9045, 1:1000), HSP70 (CY5496, 1:1000), and E-cadherin (ab40772, 1:1000) were purchased from Abways (Shanghai, China). The antibody for His-Tag (230001, 1:1000) was purchased from Zen-Bio (Chengdu, China).

For endogenous co-immunoprecipitation (Co-IP), the nucleoli were lysed in high salt RIPA buffer (50 mM Tris pH 7.5, 500 mM NaCl, 1% Nonidet P-40, 0.5% deoxycholate, and proteasome inhibitors). Cell lysates were subjected to sonication and centrifugation at $12,000 \times g$ for 30 min, and equal amounts of total protein were incubated with primary antibodies or normal indicated IgG overnight at 4 °C, and then 30 µL of protein A/G beads were added for an additional 2 h of incubation. For exogenous Co-IP, anti-HA beads (or anti-Flag beads) were added to equal amounts of total protein and incubated overnight. Beads were centrifuged ($500 \times g$ for 30 s) and washed three times using wash buffer (20 mM Tris-HCl, 250 mM NaCl, 0.2 mM EGTA, and 0.1% Nonidet P-40). The beads were heated at 100 °C for 10 min before western blot analyses. Anti-FLAG M2 affinity gel (A2220) was purchased from Sigma-Aldrich (St. Louis, USA). Pierce Anti-HA magnetic beads (#88836) were purchased from Thermo Fisher Scientific (Waltham, MA, USA).

For immunofluorescence staining, cells grown on coverslips were fixed with 4% polyformaldehyde in PBS, permeabilized with 0.1% Triton X-100 in PBS, blocked with 4% bovine serum albumin in PBS, hybridized to an appropriate primary antibody (Snail1: 1:50, sc-271977, Santa Cruz Biotechnology(CA, USA); USP36: 1:400, 14783-1-AP, Proteintech; B23: 1:50, MA5-12508, ThermoFisher; HA-Tag: 1:1000, #5017, CST), Flag-Tag: 1:1000, #8146, CST), followed by incubation with a second antibody (Goat anti-Mouse Alexa Fluor 488, A-11029 or Goat anti- Rabbit Alexa Fluor 514, A-31558, ThermoFisher). The cells were counterstained with ProLong® Gold Antifade Reagent with DAPI (#82961, CST) prior to visualization and photographed using a Leica TCS SP5II confocal laser scanning microscope. LAS X (V3.3.0) was used to analyze fluorescent images. To determine the Snail1 co-localization with nucleolar B23 or USP36, the free software Image J. Fiji coupled with the Coloc 2 plugin and Pearson's correlation coefficient were used to calculate double fluorescence correlation coefficients[53], and co-localized fluorescence quantifications were presented by scatter diagrams.

## Quantitative PCR and northern blot analyses

Quantitative PCR (qPCR) analyses were performed as described[54]. Briefly, total RNA was isolated from cells using an RNA extraction kit (Qiagen, Germany) and subjected to reverse transcription according to the manufacturer's protocol. qPCR analyses were performed in a CFX96 Real-Time PCR System (Bio-Rad) using SoFast EvaGreen Supermix (Bio-Rad). qPCR values were calculated using the ΔΔCt method. qPCR primers used in this study were listed in Supplementary Table 4.

For Northern blot analyses, total RNA was extracted using the TRIzol™ Reagent (#15596026, Invitrogen, Carlsbad, CA, Cat) according to the manufacturer's protocol. 4 µg of total RNA was resolved on 1.2% denaturing agarose gels in the presence of MOPS buffer (20 mM MOPS, 5 mM sodium acetate, and 1 mM EDTA) containing 6% formaldehyde. Agarose gels were run for 2.5 h at 75 V and then were transferred to a Hybond N+ nylon membrane (#11209272001, Roche) by capillarity overnight in 20× saline sodium citrate (SSC, 3 M NaCl +0.3 M Trisodium citrate) and fixed by UV cross-linking. Membranes were prehybridized for 1 h at 37°C in DIG-Easy Hyb buffer (#11093657910, Roche). The DIG-labeled oligo-deoxynucleotide probe was added and incubated overnight at 37 °C. The prime used in this study was listed in Supplemental Table 2. After hybridization, the membranes were washed twice for 5 min at room temperature in 2× SSC with 0.1% SDS and twice in 0.5× SSC with 0.1% SDS at 37 °C. Using blocking buffer (#11093657910, Roche) to block the non-specific antibody binding sites on the membranes. The DIG-labeled probe is

detected by high affinity anti-Digoxigenin antibodies (#ab51949, 1:1000, Abcam), coupled to HRP for chemiluminescent detection.

## Cell apoptosis and cell viability assays

For the apoptosis assay, cells were grown in 6-well plates to approximately 70% confluence prior to treatment with the appropriate chemicals for the indicated time. Cells were then incubated with Annexin V-FITC and PI (Beyotime, C1052, China) followed by flow cytometer analyses (BD FACScalibur, BD Biosciences, USA). FlowJo (v10.4.0) was used to analyze FACS data. The percentage of apoptotic cells was calculated by Annexin V+/PI+ cells. For cell viability, 3-(4,5-dimethylthiazol-2-yl)−2,5-diphenyltetrazolium bromide (MTS) assays were performed using the CellTiter 96 kit (Promega, USA).

## Chromatin immunoprecipitation (ChIP) assay

ChIP assays were performed in HCC1806 cells with ChIP-IT Kit (Active Motif, USA) using antibodies specific for HSF1 (CY9045, 1:50, Abways) or normal rabbit IgG (2729, Cell Signaling Technology). ChIP samples were subjected to PCR experiments to amplify fragments of the *USP36* gene promoter elements using indicated primers as listed in Supplementary Table 4. To examine the strength of HSF1 for binding to *USP36* gene promoter elements, ChIP samples were subjected to qPCR or reverse transcriptional PCR using primers as indicated. The value of each ChIP sample was normalized to its corresponding input.

## Mass spectrometry

HEK-293 cells stably expressing HA-Snail1 were transiently transfected with Flag-USP36 or Vector control for 48 h. Cells were treated with MG132 for 6 h followed by immunoprecipitation. For determining the ubiquitination levels in Snail1, the gel bands obtained from SDS-PAGE were sliced and destained using 50% ethanol. After fully dehydrated using 100% ACN (Acetonitrile), samples were reduced by 1.5 mg/mL DTT at 56 °C for 1 h and then alkylated by 10 mg/mL iodoacetamide in darkness for 45 min at room temperature. The gels were then dehydrated in 100% ACN and the proteins were digested by trypsin at a ratio of 1:50 (w/w, trypsin/protein) for approximately 16 h at 37 °C. The tryptic peptides were extracted sequentially in 50% ACN/5% TFA (Trifluoroacetic Acid), 75% ACN/0.1% TFA, and 100% ACN, then the extracted peptides were combined and dried by SpeedVac (Hunan Herexi) and desalted with C18 ZipTip (Millipore) before LC-MS/MS analysis.

The desalted peptides were resuspended using buffer A (2% ACN, 0.1% FA) and then loaded onto a homemade trap column (2 cm length × 75 µm inner diameter, Spursil C18 5 µm particle size, DIKMA), which was coupled to a homemade capillary column (20 cm length × 75 µm inner diameter, Reprosil-Pur C18-AQ 1.9 µm particle size). For LC-MS/MS analysis, an EASY-nanoLC 1000 nanoflow LC instrument (Thermo Fisher Scientific, San Jose, USA) was used in combination with a high-resolution mass spectrometer (Q Exactive Plus, Thermo Fisher Scientific). Peptides were separated and eluted with a gradient of 13% to 100% HPLC buffer B (0.1% formic acid in 80% acetonitrile, v/v) in buffer A (0.1% formic acid in 98% water, v/v) at a flow rate of 330 nL/min. Data-dependent acquisition (DDA) was performed in positive ion mode. Full MS was acquired in the Orbitrap mass analyzer over the range of $m/z$ 350 to 1800 with a resolution of 70,000 at $m/z$ 200. The automatic gain control (AGC) value was set at $3 \times 10^{-6}$ with a maximum injection time of 20 ms. The top 20 most intense parent ions were selected for MS/MS scans with a 1.6 $m/z$ isolation window and fragmented with a normalized collision energy (NCE) of 27%. The AGC value for MS/MS was set to a target value of $1 \times 10^{-6}$, with a maximum injection time of 64 ms and a resolution of 17,500. Parent ions with a charge state of $z = 1$ or 8 or with unassigned charge states were excluded from fragmentation, and the intensity threshold for selection was set to $3.1 \times 10^{-5}$.

The raw files obtained were searched against the Swiss-Prot human protein sequence database (updated on 01/2017; 20,413 protein sequences) by using MaxQuant (version 1.6). The precursor ion mass errors of all identified peptides were found to be within 10 ppm, and the fragment ion mass tolerance was set at 0.02 Da. Lysine ubiquitination was specified as variable modification. The minimum peptide length was set at 6 amino acids. The maximum allowed missed trypsin cleavages were set at 2, and peptides were not nested within another longer peptide. Proteins with a false discovery rate (FDR) < 1% at both the protein and peptide levels were kept. (The analyzed data of mass spectrometry was listed in Supplementary Data 1).

## Photobleaching experiments

H1299 cells stably expressing Snail1-GFP were plated on glass coverslips and grown overnight. The fluorescence recovery after photobleaching (FRAP) experiments of nucleolus were performed on a Leica TCS SP5II confocal laser-scanning microscope. (Leica Microsystem). The 488 nm laser and a ×63 oil immersion objective (1.4 NA) were used in the photobleaching experiments. The region of interest was bleached with the 488 nm laser at 90% full power and subsequent scans were taken at 2% of full power. Images were taken before bleaching and then images were acquired at 5 s intervals for at least 35 min.

## Statistics and reproducibility

GraphPad Prism 8.0 (GraphPad Software Inc., USA) was used for data recording, collection, processing, and calculation. Data from at least three independent experiments in vitro were presented as mean ± standard deviation (SD), and data from animal experiments were presented as mean ± standard error of mean (SEM). Unpaired two-tailed Student's *t* test was used for comparing two groups of data. One/two-way ANOVA with Tukey's test or Bonferroni's test was used to compare multiple groups of data. *P* values ≤ 0.05 were considered significant.

## Data availability

The mass spectrometry proteomics data generated in this study have been deposited to the ProteomeXchange Consortium via the PRIDE partner repository with the dataset identifier PXD045622 (https://www.ebi.ac.uk/pride/). All data generated or analyzed during this study are included in this article and its Supplementary Information files. The uncropped gel or blot figures and original data underlying Figs. 1–7 and Supplementary Figs. S1–S9 are provided as a Source data file. Source data are provided with this paper.

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

## Acknowledgements

We are grateful to Dr. Lingqiang Zhang (State Key Laboratory of Proteomics, National Center for Protein Sciences, Beijing Institute of Lifeomics, China.) for His-Ub, His-Ub-K48-only, and His-Ub-K63-only plasmids. We thank members of the Z.-X.J.X. laboratory for stimulating discussions during the study. This work was supported by the National Natural Science Foundation of China (81830108 and 82073248) to Z.-X.J.X. and Y.Y., National Key R&D Program of China (2022YFA1103700) to Z.-X.J.X., and Natural Science Foundation of Sichuan Province (2023NSFSC1859 and 2023NSFSC1861) to Y.Y. and M.N.

## Author contributions

Y.Y. and Z.-X.J.X. conceived and designed the research. K.Q. and S.Y. performed most of the experiments with assistance from Y.L., R.G., J.F., S.G., Y.W., K.J., Z.X., and F.L. H.C., M.N., M.-S.D., L.D., Y.C., and Y.Z. contributed to the data discussion. Y.Y. and Z.-X.J.X. wrote the manuscript. All the authors read and approved the manuscript.

## Competing interests

The authors declare no competing interests.
