## [Peer Review File · Nature Communications]

Reviewers' Comments:

Reviewer #1:

Remarks to the Author:

In this manuscript, the authors found that Snail1 can be stabilized by USP36 and localized in the nucleolus, where it induces 47S pre-rRNA expression to antagonize the ribotoxic effect from HHT. They further found that the JNK signaling pathway was responsible for USP36 induction and the consequent Snail1 stabilization in the nucleolus. Combination of HHT with either JNK inhibitor or knockdown of USP36 or Snail1 can promote apoptosis in solid tumor cells. Although some of these findings are interesting, most of the experiments are based on overexpression system. The major conclusion remains premature and requires more substantial evidence to further strengthen. Detailed comments are listed below.

1) Based on immunofluorescent staining (IF) in Fig 1A/B and western blot (WB) in Fig 2F, it seems about 20% of Snail1 was localized in the nucleolus. This could be due to the high levels of Snail1 in these tumor cells, a total cellular level of Snail1 expression in the cell lines used should be presented by WB. In addition, the percentage of nucleolus Snail1 in Fig 1A/B should be included to avoid bias of image selection.

2) In Fig 2F, MG132 treatment stabilizes Snail1 in both the nuclear plasma and nucleolus (lane 1 vs lanes 3 & 5), suggesting that the nuclear Snail1 can also be stabilized by MG132 in addition to USP36. However, HHT blocks the nucleolus translocation of K157R-Snail1 (Fig 3B), whereas MG132 does not and increases the nucleolus Snail1 expression (Fig 3J). These seem to be contradictory to each other.

3) Although cellular fractionation is used to study the nuclear-nucleolus shuttling, this may not be accurate due to contamination of cellular fractions. An interspecies heterokaryon assay is preferred to confirm unambiguously how USP36 or JNK signaling promotes the shuttle of Snail1 from one nucleolus to the other.

4) In Fig 4, K157R-Snail1 cannot induce 47S pre-rRNA levels compared with WT-Snail1, is this due to the defect of nucleolus translocation of K157R-Snail1? USP36-knockdown reduces the level of 47S pre-rRNA, and the reduction can be rescued by WT-Snail1 but not K157R-Snail1. Because only a small percentage of WT-Snail1 is localized in the nucleolus, this does not rule out the possibility of the majority of nuclear Snail1 achieving this effect.

5) In Fig 4J – Fig 4N, it seems that rescued expression of Snail1 can completely restore the effect of USP36 in promoting cell survival under HHT treatment, suggesting that other potential targets of USP36 are not important in cell survival. Have other USP36 target substrates been examined? These surrogate markers should be included.

6) In Fig 5A/B, the comparison between breast cancer cell lines (SUM159, HCC1806, or Hs 578T) and leukemia cells (K562, OCI-AML2, or MOLM-13) is not appropriate, because they have different genetic backgrounds other than USP36 and Snail1. Similarly, will expression of USP36 or Snail1 block HHT-induced apoptosis in leukemia cells? Furthermore, JNK can regulate many target substrates and its downstream transcriptional pathway, can JNK alone induce endogenous Snail1 stabilization and nucleolus translocation? Will rescued Snail1 expression block JNK inhibitor-induced apoptosis?

Reviewer #2:

Remarks to the Author:

In this manuscript, entitled "Enhancing nucleolar Snail1 stabilization by USP36 facilitates ribosome biogenesis and cancer cell survival in response to ribotoxic stress", the authors report that ribotoxic stress induces nucleolar accumulation of Snail1. This appears to be achieved through JNK pathway activation which leads to upregulation of USP36 and in turn deubiquitinates and stabilizes Snail1. The authors identify K157 of Snail1 as being essential for association with USP36. HHT, a translation inhibitor, and a drug used in the clinic towards leukemic cells, is

ineffective against human tumors. The authors show that inhibiting the JNK-USP36-Snail axis sensitizes solid tumors to HHT. Overall, this is an extremely well performed study. The data strongly support the conclusions, the author's interpretation of their data is cautious and validated by independent means where possible, the paper is well presented and an easy read. The submitted manuscript delineates a molecular mechanism that leads to Snail nucleolar accumulation and they illustrate how that mechanism is circumvented in solid tumors and provide an approach to targeting it.

I have but a few minor comments for the authors.

-For the sake of being complete, can they provide magnification bars for the photographs. I see a few scattered here and there, but a set for each row might be appropriate.

-Fig 2H, I don't see Snail1 very well in the "minus" HHT lane of the shUSP36 panel. Therefore, it's difficult to process the statement made by the authors in the text that "Silencing of USP36 significantly inhibited HHT-induced Snail1 accumulation in the nucleoli" then be made (line 143)."

-Fig 3J. Can the authors provide a set of Western blots to solidify/confirm this data.

-Line 195-196: Can the authors please reword this statement since in the previous sentence the wrote "K157 of Snail1 is not required for Snail1 into the nucleolus." Also a grammatical correction is required for the sentence spanning lines 192-194.

-If Snail1 stabilization leads to increased 47S pre-rRNA levels, does one see increased ribosome levels? Would an rRNA Northern blot be helpful here to ensure that the 47S is processed appropriated. This information would complement the nice data in Fig 4.

-Fig 4P – What levels of Snail1 knockdown is being achieved in these experiments by the shRNAs targeting Snail1?

Reviewer #3:

Remarks to the Author:

Summary

In the submitted work, the authors identify USP36 as a DUB of Snail1 and investigate how they coordinate a response to HHT-induced ribotoxic stress. The half-life of transcription factor Snail1 is short (minutes, hours?) and normally degraded via the ubiquitin proteasome pathway. The authors demonstrate that Snail1 is localized to the nucleolus following ribotoxic stress (HHT, anisomycin, puromycin, G418, blasticidin) using immunofluorescence and fractionation assays and is stabilized following HHT treatment by the nucleolar DUB USP36. Treatment with ribotoxic stress (HHT, anisomycin, blasticidin) induces Snail1 stabilization and JNK signaling. Snail1 and USP36 are shown to stably interact via co-IP, and K157 of Snail1 is important for this interaction as well as localization of Snail1 to the nucleolus. Snail1 and USP36 appear to promote ribosome biogenesis through expression of 47S pre-rRNA in response to HHT. Lastly, treatment with ribotoxic stress (HHT, anisomycin, blasticidin) and JNK signaling inhibitor SP600125 induces expression of apoptotic marker CC3 in breast cancer cell lines, whereas treatment with ribotoxic stress (HHT, anisomycin, blasticidin) alone induces expression of apoptotic marker CC3 in leukemia cell lines. There is no obviously stated hypothesis or biological question the authors are addressing. Rather, known information about Snail1 is presented and then followed by what the authors demonstrated, leaving the reader to wonder what the specific biological question or motivation was driving the research in the first place if any. It is interesting that modulating the JNK-USP36-Snail1 pathway has therapeutic potential with respect to certain cancers. However, I found it odd the authors did not further pursue how USP36 recognizes Snail1 or regulates Snail1 nucleolar localization/accumulation. Presumably this information must be known before using drugs to modulate the JNK-USP36-Snail1 pathway in therapeutic treatments.

There are some serious concerns about the rigor of experiments (majority of experiments were done without complementation of WT/mutant cDNAs in knockdown conditions or with separation-of-function mutations), data reliability (lack of quantifications), particularly IF and Western blot data, because nowhere in the manuscript does it mention replicates (biological or technical)

performed for a given experiment. Quantification and statistical analyses of IF and Western blots would allow the authors to claim data were significant but is absent. Information about how experiments were conducted is lacking. In the materials and methods section, assays and other techniques were performed as previously described, which is not in line with submission guidelines for Nature Communications nor helpful for anyone wishing to easily reproduce the author's work. The writing is disjointed and there are numerous mistakes in language and grammar throughout, suggesting the manuscript was not carefully reviewed for clarity or errors prior to submission. Overall, I think this manuscript is better suited for a more specialized journal, even after revision.

Major issues and questions:

- ♣ How does JNK signaling regulate USP36 activity?
- ♣ The initial motivation for Figure 1 should be more obviously stated. The authors "performed an IF assay to examine subcellular localization of Snail1..." without explaining why they performed this experiment in the first place. What is the question they are trying to address? Additionally, colocalization of Snail1 and B23 is observed in some cells (Fig 1A, B), not all, and a distinction I think worth noting. It would be helpful to include quantifications of B23/Snail1 colocalization with and without HHT treatment in the IF experiments in Fig 1C so the authors can confidently state HHT treatment significantly induces nucleolar localization of Snail1.
- ♣ In the discussion the authors state their results suggest K157 of Snail1 is a critical binding site for USP36. Is it not possible that the ubiquitin chain conjugated to K157 is itself critical for USP36 binding/recognition? More work needs to be done to address the role of K157 in mediating USP36 binding (direct or indirect?). More mutants need to be generated and tested to understand the significance of K157 site/region for mediating binding.
- ♣ The authors state in Fig 2A that "HHT significantly increased Snail1 protein half-life while Snail1 mRNA levels were modestly upregulated." Use of "significantly" implies statistical analysis, which is absent. Only quantification of a single CHX Western blot is shown. Were experiments repeated? The figure legend does not address the number of replicates.
- ♣ There is poor expression of OTUD5, USP37, and USP44 compared to other DUBs in Fig 2B
- ♣ Regarding Fig 2N - the text should include information on His-Ub plasmids since none is given and may not be obvious to those outside the ubiquitin field. This section should describe how giving cells ubiquitin in which only one K residue remains available for ubiquitin chain formation informs understanding of USP36 activity.
 - o Unclear whether USP36 is recognizing Snail1 or ubiquitin chains on Snail1. This could be determined by blocking ubiquitination with an E1 inhibitor then probing for interaction between USP36-C131A and Snail1.
 - o Biochemical deubiquitination assays would strengthen conclusions here. Measuring DUB activity of purified USP36 w/ purified chains of different linkage (K48, K63) would reveal USP36 linkage specificity and whether it cleaves en bloc or in a processive manner (one ubiquitin at a time). Perhaps beyond the scope of the current manuscript.

o

Minor issues and questions:

- ♣ Manuscript may flow better if experiments investigating interaction between USP36 and Snail1 were grouped in the same figure – Fig 2J, K, M, N and Fig 3
- ♣ MG132 is used in Fig 2B to stabilize HA-Snail1. I think it would be informative to readers outside of the ubiquitin-proteasome field to state that MG132 is proteasome inhibitor and thus blocks all proteasome-mediated degradation.
- ♣ Authors state Snail1 accumulates in nucleoli following ectopic expression of USP36 in Figure 2G, but nucleoli markers are lacking in the IF experiment. Later in the text they state USP36 has been shown to localize exclusively to nucleoli, which should be stated earlier when describing Fig 2G.
- ♣ Regarding Fig 2I, the text should state how stable protein complexes between USP36 and Snail1 were detected. The authors show USP36 and Snail1 interact via pulldown in presence of 500 mM NaCl. May be worth including they interacted under high salt conditions to strengthen the argument they stably/strongly associate. The stoichiometry of USP36 and Snail1 should also be determined.
- ♣ Does Snail1-K157R fail to upregulate 47S pre-rRNA expression because Snail1-K157R does not localize to the nucleolus?
- ♣ Figure 5
 - o Why is there unequal GAPDH loading in Figure 5A between breast cancer cells and leukemia cells?

- o By eye it looks like there is more T-JNK signaling activation in the leukemia cells compared to the TN breast cancer cells in Fig 5A, but this is hard to tell because there is unequal loading as observed by GAPDH bands.
- o What is the difference between T-JNK and p-JNK?
- ♣ Language and grammar
 - o Lines 42 and 43, inconsistent language - "such as" and "e.g.,"
 - ♣ "... inhibitors (such as anisomycin,..."; "...ribotoxins (e.g., ricin, Shiga toxin..."
 - o Lines 46-47, scientific name *Cephalotaxus harringtonia* should be italicized
 - o Line 80, *in vitro* and *in vivo* should be italicized
 - o Line 63, "ovarian" should be replaced with "ovaries"
 - o Word missing in line 194, "...K157 of Snail1 is not required for Snail1 [transport/localization] into the nucleolus."
 - o Word missing in line 294, "...between [the] cytoplasm and nucleoplasm."
 - o Spelling and grammar in line 340, "... shows that despite frequent mutations are found in Snail1, Lys157 of snial1 is rarely..."
 - o Confusing language lines 362-364, "Indeed, HHT is unable to activate JNK nor to upregulate the expression..."
- ♣ Figure legends should include full phrase in addition to abbreviation even if stated in earlier figure legends to help reader understand figure without referencing previous figure legends for clarification.
 - o Figure 3 includes CP, NP, and No, but does describe what they are in the figure legend.
 - o Figure 4I includes CC3 on the Western blot, but no mention of what that is in the figure legend. CC3 is listed in the materials and methods, but is described as caspase-3 in main body text. Some consistency would help the reader.
 - o Figure 5A includes T-JNK on the Western blot, which is not described anywhere in the manuscript. What is the difference between T-JNK and P-JNK? Presumably a typo since the authors are probing for dually phosphorylated p46 and p54. Some explanation in the text would help the reader.
- ♣ In the materials and methods section under Western blots, it would be nice to include antibody dilutions used.
- ♣ The ruler number markings in Figure 4P are backwards, but maybe that doesn't matter since it's still legible.

Rebuttal

Reviewer #1

Comments:

In this manuscript, the authors found that Snail1 can be stabilized by USP36 and localized in the nucleolus, where it induces 47S pre-rRNA expression to antagonize the ribotoxic effect from HHT. They further found that the JNK signaling pathway was responsible for USP36 induction and the consequent Snail1 stabilization in the nucleolus. Combination of HHT with either JNK inhibitor or knockdown of USP36 or Snail1 can promotes apoptosis in solid tumor cells. Although some of these findings are interesting, most of the experiments are based on overexpression system. The major conclusion remains premature and requires more substantial evidence to further strengthen. Detailed comments are listed below.

Response: We sincerely appreciate the reviewer's constructive comments. In our previous manuscript, we showed that USP36 knockdown leads to reduced Snail1 accumulation in the nucleolus upon ribotoxic stress (Revised Figure 2H). Moreover, silencing of USP36 suppresses Snail1 protein expression accompanied by reduced 47S pre-RNA levels (Revised Figure S2D and 4F). In addition, the knockdown of either USP36 or Snail1 facilitates HHT-induced breast cancer cell death *in vitro* and tumor growth inhibition *in vivo* (Revised Figure 4M-4P and 4R-4T).

In order to strengthen the effects of USP36 ablation on the nucleolar Snail1 accumulation and its biological significance, we performed a series of new knockdown experiments. Our results indicated that the knockdown of USP36 not only promoted Snail1 protein ubiquitination and degradation (shown below in the Rebuttal Figure 1A-1B), it also led to markedly reduced 47S pre-rRNA (Rebuttal Figure 1C). Moreover, silencing of Snail1 also markedly inhibited HHT-induced upregulation of 47S pre-RNA levels (Rebuttal Figure 1D). As another approach to USP36 inhibition, we employed SP600125, a pharmacological inhibitor of JNK, to demonstrate that SP600125 efficiently suppressed HHT-mediated upregulation of USP36 and Snail1 nucleolar accumulation (Rebuttal Figure 1E-1G), resulting in significant suppression of breast tumor growth *in vivo* (Rebuttal Figure 1H-1I).

Rebuttal Figure 1. (A-B) Silencing of USP36 promoted Snail1 protein ubiquitination (A) and degradation (B). **(C)** Knockdown of USP36 reduced 47S pre-rRNA and 34S-rRNA levels, as evidenced by Northern Blot analyses. **(D)** Silencing of Snail1 inhibited HHT-induced upregulation of 47S pre-RNA levels, as evidenced by Northern Blot analyses. **(E-G)** SP600125 efficiently suppressed HHT-mediated upregulation of USP36 and Snail1 expression (E) and Snail1 nucleolar accumulation (F-G) *in vivo*. **(H-I)** A combination of SP600125 and HHT significantly inhibited tumor growth *in vivo*.

These new results, in complementing the conclusions obtained from the overexpression systems, provide solid evidence that USP36-mediated nucleolar Snail1 accumulation is critically important in ribosome biogenesis and tumor growth in response to ribotoxic stress. **The new results were integrated in revised Figure 2K, 2M, 4I, 4K, and 5G-5K.**

To substantiate our major conclusions, we further dissected the molecular mechanisms underlying the HHT-JNK-USP36-Snail1 axis in the ribotoxic response. Results from the mass spectrum and point mutation analyses indicated that USP36 deubiquitinated Snail1 protein on K146 and K206 (Rebuttal Figure 2A-2D). Consistently, HHT significantly upregulated wild-type Snail1 expression, but not Snail1^{K146/K206} (Snail1^{2KR}) expression (Rebuttal Figure 2E). Moreover, we show that USP36 interacted with Snail1 regardless of the status of its ubiquitination (Rebuttal Figure 2F). Interestingly, ZDOCK protein-protein docking analyses suggested that the C2H2-type Zinc finger 1 (154-176 aa, ZnF1) of Snail1 protein is critical for USP36-Snail1 interaction (Rebuttal Figure 2G), which was validated by our truncation experiments (Rebuttal Figure 2H). Our previous results showed that the K157 residue of ZnF1 is critical for USP36-Snail1 protein complex formation and Snail1 protein nucleolar accumulation (Revised Figure 3F-3O). **The new results were integrated in revised Figure 2O-2U, 3D-3E, S3A, and S3C.**

Rebuttal Figure 2. (A-B) USP36 deubiquitinated Snail1 protein on K146 and K206. **(C-D)** Ectopic expression of USP36 significantly upregulated Snail1^{WT} protein expression and stability, but not Snail1^{2KR}. **(E)** HHT significantly upregulated Snail1^{WT} protein expression, but not Snail1^{2KR}. **(F)** USP36-Snail1 could form a stable protein complex even Snail1 protein ubiquitination is inhibited by E1 inhibitor TAK-243. **(G)** The C2H2-type Zinc finger 1 (154-176aa, ZnF1) of Snail1 protein is critical for USP36-Snail1 protein complex formation, as predicted by ZDOCK protein-protein docking analyses. **(H)** The ZnF1 region of the Snail1 protein is essential for USP36-Snail1 protein complex formation.

We previously showed that HHT activated JNK to stimulate USP36 transcription. We further elucidated how JNK signaling regulates USP36 expression. Our new results showed that JNK upregulated HSF1 protein expression, which in turn transactivated USP36 via direct binding on the *USP36* gene promoter -393 to -279 region (Rebuttal Figure 3A-3B). Furthermore, inhibition of JNK signaling completely inhibited HHT-induced upregulation of HSF1 and USP36 protein expression (Rebuttal Figure 3C). Importantly, the pharmacological inhibition of HSF1 by KRIBB11 effectively inhibited HHT-

mediated upregulation of USP36 and Snail1 expression (Rebuttal Figure 3D-E). **The new results were integrated in revised Figure S6G-S6M.**

Rebuttal Figure 3. (A) Overexpression of JNK1 promotes HSF1 expression. **(B)** HSF1 direct binding on the *USP36* gene promoter -393 to -279 region, as evidenced by CHIP analyses. **(C)** Inhibition of JNK signaling by SP600125 completely inhibited HHT-induced upregulation of HSF1 and USP36 protein expression. **(D)** HSF1 inhibitor KRIBB11 could markedly suppress USP36 and Snail1 protein expression. **(E)** Inhibition of HSF1 by KRIBB11 effectively inhibited HHT-mediated upregulation of USP36 and Snail1 expression.

To further support that nucleolar Snail1 promotes 47S pre-rRNA expression, we newly performed Northern blot analyses. Overexpression of wild-type Snail1 (Snail1^{WT}), but not Snail1^{K157R} which is defective in nucleolar accumulation, upregulated 47S pre-rRNA expression accompanied by increased 34S rRNA levels (Rebuttal Figure 4A). Furthermore, rescuing experiments indicated that Snail1 could totally rescue USP36 knockdown-mediated reduction of 47S pre-rRNA and 34S rRNA expression (Rebuttal Figure 4B). In addition, the knockdown of Snail1 could also rescue HHT-mediated upregulation of 47S pre-rRNA and 34S rRNA levels (Rebuttal Figure 4C). **The new results were integrated in revised Figure 4D, 4I, and 4K.**

Rebuttal Figure 4. (A) Overexpression of Snail1^{WT}, but not Snail1^{K157R}, upregulated 47S pre-rRNA and 34S rRNA expression, as evidenced by Northern blot analyses. **(B)** Snail1^{WT}, but not Snail1^{K157R}, could totally rescue USP36 knockdown-mediated reduction of 47S pre-rRNA and 34S rRNA expression. **(C)** Knockdown of Snail1 completely rescued HHT-mediated upregulation of 47S pre-rRNA and 34S rRNA expression.

Taken together, substantial new experiments were performed to address the deficits of overexpression systems and the depth of underlying molecular mechanisms. Our results provide solid evidence in supporting our conclusion that USP36-mediated nucleolar Snail1 accumulation is critically important in ribosome biogenesis and tumor growth in response to ribotoxic stress.

The following are responses to specific comments:

Point 1:

1) Based on immunofluorescent staining (IF) in Fig 1A/B and western blot (WB) in Fig 2F, it seems about 20% of Snail1 was localized in the nucleolus. This could be due to the high levels of Snail1 in these tumor cells, a total cellular level of Snail1 expression in the cell lines used should be presented by WB. In addition, the percentage of nucleolar Snail1 in Fig 1A/B should be included to bias of image selection.

Response: We thank the reviewer for the constructive suggestions. Accordingly, we performed a series of new experiments with quantification analyses. As shown in Rebuttal Figure 5, without ribotoxic stress, Snail1 protein was mainly localized in the nucleoplasm and only 4% of HCC1806 cells exhibited positive nucleolar Snail1 staining (Rebuttal Figure 5A), in which about 18% of total nuclear Snail1 protein was attributable to the nucleolus (Rebuttal Figure 5B). Upon ribotoxic stress, more than 97% of HCC1806 cells exhibited positive nucleolar Snail1 staining (Rebuttal Figure 5A). **The new analyses were integrated in revised Figure 1H and S1.**

A

Group	Total HCC1806 cells	Nucleolar Snail1 positive HCC1806 cells	Percentage of Nucleolar Snail1 positive HCC1806 cells (%)
Ctrl	197	8	4.06
HHT	197	194	98.48
Anisomycin	201	197	98.01
Puromycin	208	202	97.12
G418	216	211	97.69
Blasticidin	217	213	98.16

B

Cell number	Fluorescence intensity in nucleoli	Fluorescence intensity in nucleus	Nucleolar/Nuclear fluorescence intensity (%)	Mean (%)
#1	21.906	115.648	18.94	18.24
#2	24.92	120.161	20.74	
#3	20.579	116.95	17.60	
#4	15.286	97.532	15.67	

Rebuttal Figure 5. (A) Upon ribotoxic stress, more than 97% of HCC1086 cells exhibited positive nucleolar Snail1 staining. **(B)** About 18% of total nuclear Snail1 protein was attributable to the nucleolus in nucleolar Snail1 positive HCC1806 cells.

Our new results support the notion that high levels of Snail1 protein are important but not sufficient for Snail1 nucleolar accumulation. Results from our photobleaching experiments indicate that Snail1 proteins most likely enter the nucleolus from nucleoplasm by diffusion (Rebuttal Figure 6), in a manner similar to c-Myc, p21, and H2B nucleolar entry [J Cell Sci., 2003; PMID: 12665552; Traffic., 2010, PMID: 20331843; Biochim Biophys Acta., 2011, PMID: 21095207], implying that high levels of Snail1 protein would contribute to its nucleolar accumulation. **The new results were integrated in the revised Figure S5.**

Rebuttal Figure 6. H1299 cells stably expressing Snail1-GFP (A) were transiently transfected with USP36 for 48h (B) or were treated with the 10 μ M proteasome inhibitor MG132 for 6 h (C). Cells were subjected to photobleaching analyses.

Notably, our results provide compelling evidence that USP36 plays a critical role in Snail1 nucleolar accumulation. High levels of Snail1 through Snail1 overexpression in HCC1806 cells yield much less nucleolar Snail1 accumulation than that in HHT-treated HCC1806 cells with much less snail1 protein expression (Rebuttal Figure 7 and revised Figure 1B-1C, 1I, and 3L-3M), while the knockdown of USP36 effectively blocked HHT-induced nucleolar Snail1 accumulation (Revised Figure 2H). These results indicate that USP36, but not Snail1 levels, is essential in guiding Snail1 nucleolar accumulation.

Rebuttal Figure 7. HCC1806 cells stably overexpressed wild-type Snail1 or were

treated with HHT for 24 h. Cells were subjected to western blot analyses.

Pearson's correlation coefficient is often used to value the co-localization of proteins (Nature Communications, 2022, PMID: 36376313; Nat Cell Biol., 2023, PMID: 37037994). To ensure unbiased immunofluorescence image selection, we used Pearson's correlation coefficient analyses for the co-localization of Snail1 and nucleolar marker B23 to evaluate Snail1 nucleolar accumulation (Rebuttal Figure 8). **The new analyses were integrated in revised Figure 1C, 1E, 1G, 3C, and 5K.**

Rebuttal Figure 8. Pearson's correlation coefficient is used to value the co-localization of Snail1 and nucleolar marker B23.

Point 2:

2) In Fig 2F, MG132 treatment stabilizes Snail1 in both the nuclear plasm and nucleolus (lane 1 vs lanes 3 & 5), suggesting that the nucleus Snail1 can also be stabilized by MG132 in addition to USP36. However, HHT blocks the nucleolus translocation of K157R-Snail1 (Fig 3B), whereas MG132 does not and increases the nucleolus Snail1 expression (Fig 3J). These seems to be contradictory to each other.

Response 2: Thanks for the comments. Here, the role of K157 of Snail1 in Snail1 entering the nucleolus and protein stabilization are two major issues. Our previous data showed that nucleolar USP36 binds to and deubiquitinates Snail1 in the nucleolus. Importantly, K157 of Snail1 is critical for Snail1-USP36 protein complex formation (Revised Figure 3F), but not the site of USP36-mediated deubiquitination (Rebuttal Figure 2A-2D). Therefore, Snail1-K157R, unable to

bind to USP36, is highly unstable in the nucleolus, even under the insult of HHT, in keeping with our previous results that Snail1-K157R is stabilized in the nucleolus upon inhibition of proteasome by MG132 (Revised Figure 3N).

To further clarify whether MG132 affects Snail1-K157R protein accumulation in the nucleolus, we performed cellular fractionation assays. Our new data show that MG132 indeed promoted Snail1-K157R accumulation in the nucleolus (Rebuttal Figure 9), consistent with our previous immunofluorescence data (Revised Figure 3N). Interestingly, our photobleaching analyses showed that nucleoplasmic Snail1 diffuses to the nucleolus (Rebuttal Figure 6). These results provide solid evidence that K157 is critical for USP36 binding to and stabilization of Snail1 in the nucleolus, but not critical for Snail1 to enter the nucleolus. **The new result was integrated in the revised Figure 3O.**

Rebuttal Figure 9. HCC1806 cells stably expressing HA-Snail1^{WT} or HA-Snail1^{K157R} were treated with or without MG132 (10 μM) for 6 h. Cells were subjected to cell fractionation and western blot analyses. CP: Cytoplasm; NP: Nucleoplasm; No: Nucleoli.

Point 3:

3) Although cellular fraction is used to study the nuclear-nucleolus shuttling, this may not be accurate due to contamination of cellular fractions. An interspecies heterokaryon assay is prefer to confirm unambiguously how USP36 or JNK signaling promotes the shuttle of Snail1 from one nucleolus to the other.

Response 3: Thanks for the suggestion. The issue of nucleoplasm-nucleolus shuttling is important and, to our knowledge, numerous studies often employ photobleaching analyses to examine protein shuttling from nucleoplasm to nucleolus (Traffic, 2010, PMID: 20331843; Biochim Biophys Acta, 2011, PMID: 21095207; J Cell Sci, 2003, PMID: 12665552). We therefore performed photobleaching analyses to examine how Snail1 protein can shuttle from nucleoplasm to nucleolus. Our results showed that nucleoplasmic Snail1 relocated to the nucleolus after photobleaching, upon stabilization of Snail1 in the nucleolus in the presence of MG132 or USP36 (Rebuttal Figure 6). These results strongly suggest that nucleoplasmic Snail1 can diffuse into the nucleolus

and is stabilized by USP36, which can be upregulated by JNK signaling upon ribotoxic stress.

Point 4:

4) In Fig 4, K157R-Snail1 cannot induce 47S pre-rRNA levels compared with WT-Snail1, is this due to the defect of nucleolus translocation of K157R-Snail1? USP36-knockdown reduces the level of 47S pre-rRNA, and the reduction can be rescued by WT-Snail1 but not K157R-Snail1. Because only a small percentage of WT-Snail1 is localized in the nucleolus, this does not rule out the possibility of the majority of nuclear Snail1 achieves this effect.

Response 4: Thanks for the reviewer’s insightful comments. As we described in the response to Point #2 before, our data indicate that K157 of Snail1 is critical for USP36-Snail1 protein complex formation. Therefore, wild-type Snail1, but not Snail1-K157R, can be stabilized by USP36 in the nucleolus, which in turn upregulates 47S pre-rRNA levels.

It is reasonable to wonder whether nucleoplasmic Snail1 plays a role in 47S pre-rRNA biogenesis. However, since transcription of rDNA occurs only in the nucleolus, direct regulation of 47S pre-rRNA biogenesis has to be in the nucleolus. Our previous results indicate that nucleolar Snail1 is critically involved in 47S pre-rRNA biogenesis as determined by QPCR analyses (Revised Figure 4C, 4E, 4H, and 4J). To strengthen this conclusion, we performed Northern blot analyses. Our new results showed that silencing of USP36 significantly suppresses 47S pre-rRNA levels, which could be completely rescued by restoration of wild-type (WT) Snail1, but not Snail1-K157R that can accumulate in the nucleoplasm but not in the nucleolus (Rebuttal Figure 10). Therefore, these results indicate that nucleolar Snail1 is critical for rDNA transcription. **The new results were integrated in revised Figure 4G and 4I.**

Rebuttal Figure 10. HCC1806-shUSP36 cells stably expressing HA-Snail1^{WT} or HA-Snail1^{K157R} were subjected to cell fractionation (A) and northern blot (B) analyses. CP:

Cytoplasm; NP: Nucleoplasm; No: Nucleoli.

Point 5:

5) In Fig 4J – Fig 4N, it seems that reduced expression of Snail1 can completely restore the effect of USP36 in promoting cell survival under HHT treatment, suggesting that other potential targets of USP36 are not important in cell survival (No. bcz Partial rescue). Has other USP36 target substrates been examined? These surrogate markers should be included.

Response 5: We very much appreciate the reviewer's comments. Our results indicated that silencing of Snail1 significantly enhances HHT-induced cancer cell death, which can be completely rescued by restoration of wild-type Snail1, but not Snail1-K157R, suggesting that nucleolar Snail1 is critical in cancer cell survival upon ribotoxic stress (Revised Figure 4M-4N and S7D-S7E). Consistent with the silencing of Snail1, the knockdown of USP36 also significantly enhances HHT-induced cancer cell death, which can only be partly rescued by restoration of wild-type Snail1 (Revised Figure 4O-4P and S7F-S7G), suggesting that Snail1 is important but not the only downstream effector of USP36.

It has been shown that USP36 can stabilize nucleolar c-Myc involved in rDNA transcription and ribosomal biogenesis (PNAS, 2015, PMID: 25775507). However, our new results showed that HHT treatment upregulated USP36 expression while it significantly downregulated c-Myc expression (Rebuttal Figure 11), indicating that c-Myc is unlikely to contribute to HHT-mediated rDNA transcription and ribosomal biogenesis in our system. Notably, HHT has been shown to reduce c-Myc expression through suppression of NF- κ B signaling (PNAS, 2019, PMID: 30659143).

Moreover, it has been shown that USP36 regulates the stability of the nucleolar B23 (J Cell Sci., 2009, PMID: 19208757). Yet, our previous cellular fractionation analyses showed that HHT did not affect B23 protein expression in the nucleolus (Revised Figure 1H and 3M). Therefore, it is plausible that the USP36-Snail1 axis plays a critical role in the cellular response to HHT, while other factors downstream of USP36 could contribute as well. **The new result was integrated in the revised Figure S9.**

Rebuttal Figure 11. HCC1806 cells were treated with or without 20 ng/mL HHT for 24 h followed by western blot analyses.

Point 6:

6) In Fig 5A/B, the comparison between breast cancer cell lines (SUM159, HCC1806, or Hs 578T) and leukemia cells (K562, OCI-AML2, or MOLM-13) is not appropriate, because have different genetic background other than USP36 and Snail1. Similarly, will expression of USP36 or Snail1 block HHT-induced apoptosis in leukemia cells? Furthermore, JNK can regulate many target substrates and its downstream transcriptional pathway, can JNK alone induce endogenous Snail1 stabilization and nucleolus translocation? Will rescued Snail1 expression block JNK inhibitor-induced apoptosis?

Response 5: Thanks for the insightful comments and suggestions. Currently, HHT is the only ribosome inhibitor widely used for the clinical treatment of leukemia (J Hematol Oncol., 2014, PMID: 24387717). However, it has been documented that HHT exhibits little anticancer activity on solid tumors (Cancer, 2001, PMID: 11745238). To address this interesting clinical observation, we studied the underlying molecular mechanisms. One key finding from this study is that HHT is unable to activate JNK nor to upregulate the expression of USP36 and Snail1 in leukemia cells, leading to massive apoptosis (Revised Figure 5A-5B and S8A-S8B). By contrast, HHT strongly activates the JNK-USP36-Snail1 pathway that serves as a survival mechanism in solid tumor cells (Revised Figure 5A-5B and S8A-S8B). This notion is strongly substantiated by the results that the combination of HHT with a JNK inhibitor SP600125 synergistically induced apoptosis of solid tumor cells *in vitro* and inhibited xenograft solid tumor growth *in vivo* (Revised Figure 5C-5D, S8C-S8D, and Rebuttal Figure 1H-1I). Therefore, our results provide a plausible explanation that why HHT exhibits little anticancer activity on solid tumors and highlight that targeting the JNK-USP36-Snail1 pathway may be a potential strategy to overcome solid tumor resistance to ribotoxic stress.

To address whether the expression of USP36 could block HHT-induced apoptosis in leukemia cells, we performed new experiments. Our results showed that overexpression of USP36 had little effect on Snail1 expression and was unable to block HHT-induced cell death in leukemia cells (Rebuttal Figure 12A-12B), the reasons for which are currently not clear.

Rebuttal Figure 12. K562 cells stably expressing USP36^{WT} or USP36^{C131A} were subjected to western blot analyses (A). K562-USP36^{WT} or USP36^{C131A} were treated with or without 20 ng/mL HHT for 48 h. Cells were subjected to trypan blue exclusion assays (B).

To address whether JNK alone can induce endogenous Snail1 stabilization, we ectopically expressed Flag-JNK1. Our results showed that ectopic expression of JNK1 markedly upregulated USP36 and Snail1 expression (Rebuttal Figure 13), strongly supporting that JNK signaling plays a critical role in the HHT-mediated activation of the nucleolar USP36-Snail1 axis. **The new result was integrated in the revised Figure S6J.**

Rebuttal Figure 13. HEK-293 cells were transiently transfected with Flag-JNK1 for 48h followed by western blot analyses.

The question that whether Snail1 expression could block JNK inhibitor-induced apoptosis is interesting. In our experimental settings, JNK inhibitor

SP600125, alone, did not induce cancer cell apoptosis (Revised Figure 4Q and S7H). Therefore, we did not perform Snail1 overexpression experiments. However, we showed that the combination of HHT with SP600125 synergistically induces apoptosis of solid tumor cells, which can be largely rescued by overexpression of USP36 (Revised Figure 4Q and S7H). Moreover, silencing of USP36 or Snail1 also significantly facilitates HHT-induced cancer cell death (Revised Figure 4M-4P and S7D-S7G). Therefore, our results indicate that the activation of the JNK-USP36-Snail1 axis renders cancer cells resistant to ribotoxic stress.

Reviewer #2

Comments:

In this manuscript, entitled “Enhancing nucleolar Snail1 1 stabilization by USP36 facilitates ribosome biogenesis and cancer cell survival in response to ribotoxic stress”, the authors report that ribotoxic stress induces nucleolar accumulation of Snail1. This appears to be achieved through JNK pathway activation which leads to upregulation of USP36 and in turn deubiquitinates and stabilizes Snail1. The authors identify K157 of Snail 1 as being essential for association with USP36. HHT, a translation inhibitor, and a drug used in the clinical towards leukemic cells, is ineffective against human tumors. The authors show that inhibiting the JNK-USP36-Snail axis sensitizes solid tumors to HHT. Overall, this is an extremely well performed study. The data strongly support the conclusions, the author’s interpretation of their data is cautious and validated by independent means where possible, the paper is well presented and an easy read. The submitted manuscript delineates a molecular mechanism that leads to Snail nucleolar accumulation and they illustrate how that mechanism is circumvented in solid tumors and provide an approach to targeting it.

I have but a few minor comments for the authors.

Point 1:

-For the sake of being complete, can they provide magnification bars for the photographs. I see a few scattered here and there, but a set for each row might be appropriate.

Response 1: We sincerely appreciate the reviewer’s comments and suggestions. As suggested, we have made the scale bars more clear and more visible in the revised manuscript.

Point 2:

-Fig 2H, I don’t see Snail1 very well in the “minus” HHT lane of the shUSP36 panel. Therefore, it’s difficult to process the statement made by the authors in the text that “Silencing of USP36 significantly inhibited HHT-induced Snail1 accumulation in the nucleoli” then be made (line 143).”

Response 2: We apologize for the unclear immunofluorescence images. We have improved the brightness of the immunofluorescence images (Rebuttal Figure 14). Our data indicated that silencing of USP36 dramatically inhibited HHT-induced Snail1 accumulation in the nucleolus. **The adjusted result was integrated in the revised Figure 2H.**

Rebuttal Figure 14. HCC1806 cells stably expressing shUSP36 or shGFP were treated with or without 20 ng/mL HHT for 24 h. Cells were subjected to immunofluorescence staining analyses. Scale bar, 25 μ m.

Point 3:

-Fig 3J. Can the authors provide a set of Western blots to solidify/confirm this data.

Response 3: Thanks for the suggestion. Accordingly, we performed cellular fractionation and western blot analyses. We showed that proteasome inhibitor MG132 treatment significantly promoted both wild-type Snail1 and Snail1-K157R protein accumulation in the nucleolus (Rebuttal Figure 15), consistent with the immunofluorescence data (Revised Figure 3N). **The new result was integrated in the revised Figure 3O.**

Rebuttal Figure 15. HCC1806 cells stably expressing HA-Snail1^{WT} or HA-Snail1^{K157R} were treated with or without MG132 (10 μ M) for 6 h. Cells were subjected to cell fractionation and western blot analyses. CP: Cytoplasm; NP: Nucleoplasm; No: Nucleoli.

Point 4:

-Line 195-196: Can the authors please reword this statement since in the previous sentence the wrote “K157 of Snail1 is not required for Snail1 into the nucleolus.” Also a grammatical correction is required for the sentence spanning lines 192-194.

Response 4: We apologize for the unclear description. Here, the key issue is the role of K157 of Snail1 in entering the nucleolus or protein stabilization in the nucleolus. We performed photobleaching analyses and found that nucleoplasmic Snail1 diffuses to the nucleolus (Rebuttal Figure 16). In addition, cellular fractionation and immunofluorescence analyses showed that proteasome inhibitor MG132 treatment facilitates Snail1-K157R accumulation in the nucleolus (Rebuttal Figure 15 and Revised Figure 3N), suggesting that K157 is not essential for Snail1 to enter the nucleolus. Together, our results indicate that nucleoplasmic Snail1 diffuses to the nucleolus and binds to USP36, which stabilizes the Snail1 protein. K157 of Snail1 is a critical residue that determines USP36-Snail1 interaction in the nucleolus. We have rewritten this part in our revised manuscript (Lines 245-258).

Rebuttal Figure 16. H1299 cells stably expressing Snail1-GFP (A) were transiently transfected with USP36 for 48h (B) or were treated with the 10 μ M proteasome inhibitor MG132 for 6 h (C). Cells were subjected to photobleaching analyses.

Point 5:

-If Snail1 stabilization leads to increased 47S pre-rRNA levels, does one see increased ribosome levels? Would an rRNA Northern blot be helpful here to ensure that the 47S is processed appropriately. This information would complement the nice data in Fig 4.

Response 5: We very much appreciated the reviewer's comment and suggestion. As suggested, we have performed Northern blot analyses and found that wild-type Snail1, but not Snail1-K157R which is defective in nucleolar accumulation, upregulated 47S pre-rRNA expression accompanied by increased 34S rRNA levels (Rebuttal Figure 17A). Furthermore, rescuing experiments indicated that Snail1 could totally rescue USP36 knockdown-mediated reduction of 47S pre-rRNA and 34S rRNA expression (Rebuttal Figure 17B). Moreover, the knockdown of Snail1 could also rescue HHT-mediated upregulation of 47S pre-rRNA and 34S rRNA levels (Rebuttal Figure 17C). Together, these results further demonstrate that nucleolar USP36-Snail1 plays a critical role in regulating rDNA transcription and ribosome biogenesis. **The new results were integrated in revised Figure 4D, 4I, and 4K.**

Rebuttal Figure 17. (A) Overexpression of Snail1^{WT}, but not Snail1^{K157R}, upregulated 47S pre-rRNA and 34S rRNA expression, as evidenced by Northern blot analyses. **(B)** Snail1^{WT}, but not Snail1^{K157R}, could totally rescue USP36 knockdown-mediated reduction of 47S pre-rRNA and 34S rRNA expression. **(C)** Knockdown of Snail1 completely rescued HHT-mediated upregulation of 47S pre-rRNA and 34S rRNA expression.

Point 6:

-Fig 4P – What levels of Snail1 knockdown is being achieved in these experiments by the shRNAs targeting Snail1?

Response 6: The knockdown of Snail1 led to approximately 20% of the controlled samples (Revised Figure 4M), which were then used in the xenograft experiments (Previous Figure 4P/Revised Figure 4R).

Reviewer #3

Comments:

Summary

In the submitted work, the authors identify USP36 as a DUB of Snail1 and investigate how they coordinate a response to HHT-induced ribotoxic stress. The half-life of transcription factor Snail1 is short (minutes, hours?) and normally degraded via the ubiquitin proteasome pathway. The authors demonstrate that Snail1 is localized to the nucleolus following ribotoxic stress (HHT, anisomycin, puromycin, G418, blasticidin) using immunofluorescence and fractionation assays and is stabilized following HHT treatment by the nucleolar DUB USP36. Treatment with ribotoxic stress (HHT, anisomycin, blasticidin) induces Snail1 stabilization and JNK signaling. Snail1 and USP36 are shown to stably interact via co-IP, and K157 of Snail1 is important for this interaction as well as localization of Snail1 to the nucleolus. Snail1 and USP36 appear to promote ribosome biogenesis through expression of 47S pre-rRNA in response to HHT. Lastly, treatment with ribotoxic stress (HHT, anisomycin, blasticidin) and JNK signaling inhibitor SP600125 induces expression of apoptotic marker CC3 in breast cancer cell lines, whereas treatment with ribotoxic stress (HHT, anisomycin, blasticidin) alone induces expression of apoptotic marker CC3 in leukemia cell lines.

There is no obviously stated hypothesis or biological question the authors are addressing. Rather, known information about Snail1 is presented and then followed by what the authors demonstrated, leaving the reader to wonder what the specific biological question or motivation was driving the research in the first place if any. It is interesting that modulating the JNK-USP36-Snail1 pathway has therapeutic potential with respect to certain cancers. However, I found it odd the authors did not further pursue how USP36 recognizes Snail1 or regulates Snail1 nucleolar localization/accumulation. Presumably this information must be known before using drugs to modulate the JNK-USP36-Snail1 pathway in therapeutic treatments.

There are some serious concerns about the rigor of experiments (majority of experiments were done without complementation of WT/mutant cDNAs in knockdown conditions or with separation-of-function mutations), data reliability (lack of quantifications), particularly IF and Western blot data, because nowhere in the manuscript does it mention replicates (biological or technical) performed for a given experiment. Quantification and statistical analyses of IF and Western blots would allow the authors to claim data were significant but is absent. Information about how experiments were conducted is lacking. In the materials and methods section, assays and other techniques were performed as previously described, which is not in line with submission guidelines for Nature Communications nor helpful for anyone wishing to easily reproduce the author's work. The writing is disjointed and there are numerous mistakes in language and grammar throughout, suggesting the manuscript was not carefully reviewed for clarity or errors prior to submission. Overall, I think this manuscript is better suited for a more specialized journal, even after revision.

Response: We very much appreciated the reviewer's comments and suggestions.

Major issues #1: The biological question of this study.

Response: Elevated ribosome biogenesis is a hallmark of cancer cells. Snail1 is a key nuclear transcription factor in the regulation of epithelial-to-mesenchymal transition (EMT) and cancer metastasis. Recent studies implicate that Snail1 possesses EMT-independent functions. However, whether Snail1 localizes in the nucleolus and regulates ribosome biogenesis remains unclear. In the early stage of this study, we found that snail1 can be found in the nucleolus. Therefore, the scientific questions of this study were (1) what is the biological function of nucleolar Snail1; (2) what upstream signaling promotes nucleolar Snail1 accumulation and (3) what is the clinical significance of nuclear Snail1?

Major issue #2: mechanistic insight of USP36-Snail1 interaction and regulation.

Response: It is indeed critical to illustrate the molecular mechanisms. We first performed a series of new experiments including mass spectrum and point mutation analyses to identify the amino acid residues of Snail1 that are impacted by USP36.

Our mass spectrum analyses identified K146 and K206 of Snail1 as two amino acid residues that were deubiquitinated by USP36 (Rebuttal Figure 18A-18D). We further performed validation experiments by constructing several point/double mutations, including K146R, K206R, and K146R/K206R (2KR). We show that USP36 markedly reduced Snail1^{WT} protein ubiquitination and improved Snail1^{WT} protein stability, but not Snail1^{2KR} (Rebuttal Figure 18B-18D). In addition, HHT significantly upregulated Snail1^{WT} expression, but not Snail1^{2KR} expression (Rebuttal Figure 18E). Our results lead us to conclude that USP36 is a bona fide nucleolar deubiquitinase of Snail1 and K146/K206 on Snail1 are two key deubiquitination sites for USP36.

To identify key residues in the C-terminus of Snail1 for stable protein complex formation with USP36, we employed ZDOCK protein-protein docking analyses that predicted that the K157 in the C2H2-type Zinc finger 1 (154-176aa, ZnF1) of Snail1 protein is critical for its interaction with USP36 (Rebuttal Figure 18G), which were validated by the experiments using truncational mutant proteins (Rebuttal Figure 18H). Further experiments showed that the K157 residue of Snail1 was indispensable for its interaction with USP36 (Revised Figure 3F-3O). In addition, we show that USP36 could interact with Snail1 even in the absence of Snail1 ubiquitination (Rebuttal Figure 18F). **The new results were integrated in revised Figure 20-2U, 3D-3E, and S3C.**

Rebuttal Figure 18. (A-B) USP36 deubiquitinated Snail1 protein on K146 and K206. **(C-D)** Ectopic expression of USP36 significantly upregulated Snail1^{WT} protein expression and stability, but not Snail1^{2KR}. **(E)** HHT significantly upregulated Snail1^{WT} protein expression, but not Snail1^{2KR}. **(F)** USP36-Snail1 could form a stable protein complex even Snail1 protein ubiquitination is inhibited by E1 inhibitor TAK-243. **(G)** The C2H2-type Zinc finger 1 (154-176aa, ZnF1) of Snail1 protein is critical for USP36-Snail1 protein complex formation, as predicted by ZDOCK protein-protein docking analyses. **(H)** The ZnF1 region of the Snail1 protein is essential for USP36-Snail1 protein complex formation.

To understand how Snail1 enters the nucleolus mechanistically, we performed photobleaching analyses, which were often used to address the nucleolar translocation [J Cell Sci., 2003; PMID: 12665552; Traffic., 2010, PMID: 20331843; Biochim Biophys Acta., 2011, PMID: 21095207]. Our results showed that nucleoplasmic Snail1 could diffuse to the nucleolus after photobleaching (Rebuttal Figure 19A-19C), in a manner similar to c-Myc, p21, and H2B nucleolar entry [J Cell Sci., 2003; PMID: 12665552; Traffic., 2010, PMID: 20331843; Biochim Biophys Acta., 2011, PMID: 21095207]. Notably, the Snail1-K157R mutant, which can't bind to USP36, can accumulate in the nucleolus

after proteasome inhibitor MG132 treatment (Rebuttal Figure 19D), suggesting that although USP36 is dispensable for Snail1 nucleolar entry, it is essential for Snail1 protein stabilization and accumulation in the nucleolus.

Rebuttal Figure 19. H1299 cells stably expressing Snail1-GFP (A) were transiently transfected with USP36 for 48h (B) or were treated with the 10 μ M proteasome inhibitor MG132 for 6 h (C). Cells were subjected to photobleaching analyses. (D) HCC1806 cells stably expressing HA-Snail1^{WT} or HA-Snail1^{K157R} were treated with or without MG132 (10 μ M) for 6 h. Cells were subjected to cell fractionation and western blot analyses. CP: Cytoplasm; NP: Nucleoplasm; No: Nucleoli. Scale bar, 25 μ m.

We further addressed the mechanistic insight as to how JNK signaling regulates USP36 expression. Our new results showed that JNK upregulated HSF1 protein expression, which in turn transactivated USP36 via direct binding on the *USP36* gene promoter -393 to -279 region (Rebuttal Figure 20A-20B). Furthermore, inhibition of JNK signaling completely inhibited HHT-induced upregulation of HSF1 and USP36 protein expression (Rebuttal Figure 20C). Importantly, the pharmacological inhibition of HSF1 effectively inhibited HHT-mediated upregulation of USP36/Snail1 expression (Rebuttal Figure 20D-20E). **The new results were integrated in revised Figure S6G-S6M.**

Rebuttal Figure 20. (A) Overexpression of JNK1 promotes HSF1 expression. **(B)** HSF1 direct binding on the *USP36* gene promoter -393 to -279 region, as evidenced by CHIP analyses. **(C)** Inhibition of JNK signaling by SP600125 completely inhibited HHT-induced upregulation of HSF1 and USP36 protein expression. **(D)** HSF1 inhibitor KRIBB11 could markedly suppress USP36 and Snail1 protein expression. **(E)** Inhibition of HSF1 by KRIBB11 effectively inhibited HHT-mediated upregulation of USP36 and Snail1 expression.

Together, this revised study indicates that USP36 is a deubiquitinase of nucleolar Snail1 protein, which deubiquitinates Snail1 protein on K146 and K206 in a K48-dependent manner to promote Snail1 stabilization and accumulation in the nucleolus. The K157 residue of Snail1 protein is essential for USP36-Snail1 protein complex formation. Activation of JNK by ribotoxic stress, such as HHT treatment, promotes HSF1 expression, which upregulates *USP36* gene transcription. Nucleoplasmic Snail1 diffuses to the nucleolus and binds to USP36, which stabilizes the Snail1 protein to promote ribosome biogenesis and cancer cell survival in response to ribotoxic stress.

Major issue #3: lack of complementation experiments; inadequate quantitation analyses; missing details of methods; language and

grammar errors.

Response: We appreciate the reviewer for comments and suggestions. Accordingly, we newly performed complementation experiments and found that restoration of wild-type USP36, but not USP36-C131A, completely rescued Snail1 expression which was inhibited by silencing of endogenous USP36 (Rebuttal Figure 21). Our previous results showed that silencing of Snail1 significantly enhanced HHT-induced cancer cell death *in vitro* and tumor growth *in vivo*, which can be completely rescued by restoration of wild-type Snail1, but not Snail1-K157R (Revised Figure 4M-4N, S7D-S7E, and 4R-4T). In addition, the knockdown of USP36 also significantly enhances HHT-induced cancer cell death, however, which can also be significantly rescued by restoration of wild-type Snail1, but not Snail1-K157R (Revised Figure 4O-4P and S7F-S7G). Together, we believe that these results derived from the complementation experiments together with our results from rescuing experiments have vigorously demonstrated that the USP36-Snail1 axis plays a critical role in the suppression of HHT-mediated cancer cell death. **The new result was integrated in the revised Figure 2F.**

Rebuttal Figure 21. HCC1806-shUSP36 cells stably expressing USP36^{WT} or USP36^{C131A} were subjected to western blot analyses.

Regarding the quantitative analyses, we quantified the most of data during the revision.

We apologize for the missing materials/Methods and language/grammar errors. Accordingly, we replaced the phrase “as previously described” with a detailed description in the revised “materials and methods” section. In addition, we have carefully proofread the manuscript to minimize errors.

Specific issues and questions:

Point 1:

How does JNK signaling regulate USP36 activity?

Response 1: Thanks for the question. It has been reported that JNK can regulate

a subset of downstream transcription factors, including c-JUN, p53, YAP1, and HSF1(heat shock factor 1), to impact gene transcription. We analyzed clinical relevance and found a positive correlation in mRNA expression between HSF1 and USP36 (Rebuttal Figure 22). Our new experimental data showed that activation of JNK upregulated HSF1 expression, which in turn transactivated USP36 via direct binding on the *USP36* gene promoter -393 to -279 region (Rebuttal Figure 20A-20B). Furthermore, inhibition of JNK signaling completely inhibited HHT-induced upregulation of HSF1 and USP36 protein expression (Rebuttal Figure 20C). Importantly, the pharmacological inhibition of HSF1 effectively by KRIBB11 inhibited HHT-mediated upregulation of USP36 and Snail1 expression (Rebuttal Figure 20D-20E). **The new results were integrated in revised Figure S6F-S6M.**

Rebuttal Figure 22. The GEPIA 2 database (<http://gepia2.cancer-pku.cn>) was used to analyze the clinical relevance between JUN, FOS, TP53, YAP1, or HSF1 and USP36 in mRNA levels in breast cancers.

Point 2:

The initial motivation for Figure 1 should be more obviously stated. The authors “performed an IF assay to examine subcellular localization of Snail1...” without explaining why they performed this experiment in the first place. What is the question they are trying to address? Additionally, colocalization of Snail1 and B23 is observed in some cells (Fig 1A, B), not all, and a distinction I think worth noting. It would be helpful to include quantifications of B23/Snail1 colocalization with and without HHT treatment in the IF experiments in Fig 1C so the authors can confidently state HHT treatment significantly induces nucleolar localization of Snail1.

Response 2: We very much appreciated the reviewer's comments and constructive suggestions. Elevated ribosome biogenesis is a hallmark of cancer cells. Snail1 is a key nuclear transcription factor in the regulation of epithelial-to-mesenchymal transition (EMT) and cancer metastasis. Recent studies implicate that Snail1 possesses EMT-independent functions. However, whether Snail1 localizes in the nucleolus and regulates ribosome biogenesis remains unclear. In the early stage of this study, we found that snail1 can be found in the nucleolus Therefore, the scientific questions of this study were (1) what is the biological function of nucleolar Snail1; (2) what upstream signaling promotes nucleolar Snail1 accumulation and (3) what is the clinical significance of nucleolar Snail1? We have integrated these points in the revised manuscripts (lines 106-109 and lines 121-123).

We agree that it is important to quantify B23/Snail1 colocalization with and without HHT treatment. Pearson's correlation coefficient analyses are often used to value the colocalization of proteins (Nature Communications, 2022, PMID: 36376313; Nat Cell Biol., 2023, PMID: 37037994). We therefore used Pearson's correlation coefficient analyses for the colocalization of Snail1 and nucleolar marker B23 to evaluate Snail1 nucleolar accumulation (Rebuttal Figure 23). The quantification data showed that ribotoxic stress significantly induces nucleolar localization of the Snail1 protein. In the absence of ribotoxic stress, Snail1 protein was mainly localized in the nucleoplasm with only 4% of cells exhibiting positive nucleolar Snail1 staining. Upon ribotoxic stress, more than 97% of cells exhibited positive nucleolar Snail1 staining (Rebuttal Figure 24). **The new analyses were integrated in revised Figure 1C, 1E, 1G-1H, 3C, and 5K.**

Rebuttal Figure 23. Pearson's correlation coefficient is used to value the colocalization of Snail1 and nucleolar marker B23.

Group	Total cells	Positive cells	Percentage of positive cells (%)
Ctrl	197	8	4.06
HHT	197	194	98.48
Anisomycin	201	197	98.01
Puromycin	208	202	97.12
G418	216	211	97.69
Blasticidin	217	213	98.16

Rebuttal Figure 24. Upon ribotoxic stress, more than 97% of HCC1086 cells exhibited positive nucleolar Snail1 staining.

Point 3:

In the discussion the authors state their results suggest K157 of Snail1 is a critical binding site for USP36. Is it not possible that the ubiquitin chain conjugated to K157 is itself critical for USP36 binding/recognition? More work needs to be done to address the role of K157 in mediating USP36 binding (direct or indirect?). More mutants need to be generated and tested to understand the significance of K157 site/region for mediating binding.

Response 3: Thanks for the comments and suggestions. To address whether the ubiquitin chain conjugated to K157 contributes to USP36 binding/recognition, we performed mass spectrum analyses, which identified K146 and K206, but not K157, as amino acid residues conjugated with ubiquitin chain (Rebuttal Figure 18A-18E). In addition, we show that USP36 interacted with Snail1 regardless of the status of its ubiquitination (Rebuttal Figure 18F). Therefore, it is highly unlikely that K157 is conjugated to the ubiquitin chain for Snail1 interaction with USP36.

Then, our ZDOCK protein-protein docking analyses predicted that the K157 in the C2H2-type Zinc finger 1 (154-176aa, ZnF1) of Snail1 protein is critical for its interaction with USP36 (Rebuttal Figure 18G), which were validated by the experiments using truncational mutant proteins (Rebuttal Figure 18H). Further experiments showed that the K157 residue of Snail1 was indispensable for its interaction with USP36 (Revised Figure 3F-3O), most likely through direct interaction, as supported by the observation that USP36 can form a stable protein complex with Snail1 even in the presence of 500 mM NaCl (Revised Figure 2I and 3F).

The new results were integrated in revised Figure 2O-2U, 3D-3E, S3A, and S3C.

Point 4:

The authors state in Fig 2A that “HHT significantly increased Snail1 protein half-life while Snail1 mRNA levels were modestly upregulated.” Use of “significantly” implies statistical analysis, which is absent. Only quantification of a single CHX Western blot is shown. Were experiments repeated? The figure legend does not address the number of replicates.

Response 4: Thanks for pointing out this issue. We independently repeated the CHX experiments three times, which were quantified and statistically analyzed (Rebuttal Figure 25). This information was added in the revised Figure Legend lines 990-993.

Rebuttal Figure 25. HHT stabilizes Snail1 protein.

Point 5:

There is poor expression of OTUD5, USP37, and USP44 compared to other DUBs in Fig 2B.

Response 5: Indeed, we tried more than 3 times and found that OTUD5, USP37, or USP44 has poor expression, the reason for which is unclear. However, our new experiment results showed that ectopic expression of OTUD5, USP37, or USP44 could not upregulate Snail1 expression (Rebuttal Figure 26). **The new results were integrated in the revised Figure S2B.**

Rebuttal Figure 26. HEK-293 cells expressing HA-Snail1 were transiently transfected with USP36, USP37, USP44, or OTUD5 for 48h. Cells were subjected to Western blot analyses.

Point 6:

Regarding Fig 2N - the text should include information on His-Ub plasmids since none is given and may not be obvious to those outside the ubiquitin field. This section should describe how giving cells ubiquitin in which only one K residue remains available for ubiquitin chain formation informs understanding of USP36 activity.

Response 6: We apologize for the unclear description. We have clarified this issue in the revised manuscript (lines 970-971).

Point 7:

o Unclear whether USP36 is recognizing Snail1 or ubiquitin chains on Snail1. This could

be determined by blocking ubiquitination with an E1 inhibitor then probing for interaction between USP36-C131A and Snail1.

Response 7: As suggested, we used an E1 inhibitor TAK-243 to examine whether USP36 interacting with Snail1 depends on ubiquitin chains. As shown in Rebuttal Figure 27, inhibition of ubiquitination by TAK-243 had little effect on Snail1 interaction with wild-type USP36 or USP36-C131A, indicating the ubiquitination modification is not required for USP36 interaction with Snail1. **The new result was integrated in the revised Figure S3C.**

Rebuttal Figure 27. HEK-293 cells stably expressing HA-Snail1 were transiently transfected with Flag-USP36 (WT or C131A) for 48 h. Cells were treated with or without E1 inhibitor TAK-243 (500 nM) for 8 h prior to MG132 (10 μ M) treatment for 6 h followed by immunoprecipitation and western blot analyses.

Point 8:

o Biochemical deubiquitination assays would strengthen conclusions here. Measuring DUB activity of purified USP36 w/ purified chains of different linkage (K48, K63) would reveal USP36 linkage specificity and whether it cleaves en bloc or in a processive manner (one ubiquitin at a time). Perhaps beyond the scope of the current manuscript.

Response 8: Thanks for the comment. We agree that this important issue is beyond the scope of this study.

Minor issues and questions:

Point 9:

Manuscript may flow better if experiments investigating interaction between USP36 and

Snail1 were grouped in the same figure – Fig 2J, K, M, N and Fig 3

Response 9: Thanks for the suggestions. As suggested, we have rearranged the Figures (Revised Figure 3A-3B), which makes the presentation of this study better.

Point 10:

MG132 is used in Fig 2B to stabilize HA-Snail1. I think it would be informative to readers outside of the ubiquitin-proteasome field to state that MG132 is proteasome inhibitor and thus blocks all proteasome-mediated degradation.

Response 10: Thanks for the suggestion. We provided more information about MG132 in the revised manuscript (lines 154-155).

Point 11:

Authors state Snail1 accumulates in nucleoli following ectopic expression of USP36 in Figure 2G, but nucleoli markers are lacking in the IF experiment. Later in the text they state USP36 has been shown to localize exclusively to nucleoli, which should be stated earlier when describing Fig 2G.

Response 11: Thanks for you point out. As suggested, we described it earlier in the revised manuscript (lines 151-152).

Point 12:

Regarding Fig 2I, the text should state how stable protein complexes between USP36 and Snail1 were detected. The authors show USP36 and Snail1 interact via pulldown in presence of 500 mM NaCl. May be worth including they interacted under high salt conditions to strengthen the argument they stably/strongly associate. The stoichiometry of USP36 and Snail1 should also be determined.

Response 12: We appreciate the constructive suggestion. We rewrote to emphasize that USP36-Snail1 forms stable protein complexes even at 500 mM NaCl in the revised manuscript line 178.

Point 13:

Does Snail1-K157R fail to upregulate 47S pre-rRNA expression because Snail1-K157R does not localize to the nucleolus?

Response 13: Our results indicate that Snail1-K157R can not bind to USP36 resulting in unstable Snail1-K157R in the nucleolus and unable to upregulate 47S pre-rRNA expression.

Point 14:

Figure 5

o Why is there unequal GAPDH loading in Figure 5A between breast cancer cells and leukemia cells?

Response 14: Thanks for the comment. In our previous experiment, we equally loaded total protein derived from breast cancer cells (SUM159, HCC1806, or Hs 578T) and leukemia cells (K562, OCI-AML2, or MOLM-13). We then loaded equal levels of GAPDH protein. The result showed that HHT can activate the JNK-USP36-Snail1 pathway in breast cancer cells, but not in leukemia cells (Rebuttal Figure 28). **The new result was integrated in the revised Figure 5A.**

Rebuttal Figure 28. Triple-negative breast cancer (SUM159, Hs 578T, and HCC1806) and non-lymphocytic leukemia (K562, OCI-AML2, and MOLM-13) cells were treated with or without 20 ng/mL HHT for 24 h. Cells were subjected to western blot analyses.

Point 15:

o By eye it looks like there is more T-JNK signaling activation in the leukemia cells compared to the TN breast cancer cells in Fig 5A, but this is hard to tell because there is unequal loading as observed by GAPDH bands.

Response 14: Thanks for the comment. We have repeated this western blotting experiment and loaded equal levels of GAPDH protein. The result showed that breast cancer cell lines have comparable T-JNK levels with leukemia cells (Rebuttal Figure 28).

Point 15:

o What is the difference between T-JNK and p-JNK?

Response 15: We apologize for the unclear abbreviations. T-JNK stands for total JNK. p-JNK stands for phospho-JNK (Thr183/Tyr185). We added this information in the revised manuscript (line 939 and line 1123).

Language and grammar

Point 16:

- o Lines 42 and 43, inconsistent language - “such as” and “e.g.,”
“... inhibitors (such as anisomycin,...”; “...ribotoxins (e.g., ricin, Shiga toxin...”
- o Lines 46-47, scientific name *Cephalotaxus harringtonia* should be italicized
- o Line 80, *in vitro* and *in vivo* should be italicized
- o Line 63, “ovarian” should be replaced with “ovaries”

Response 16: Thanks for pointing out these issues. Accordingly, we have replaced them in the revised manuscript lines 63-65, lines 68-69, lines 98-99, respectively.

Point 17:

- o Word missing in line 194, “...K157 of Snail1 is not required for Snail1 [transport/localization] into the nucleolus.”
- o Word missing in line 294, “...between [the] cytoplasm and nucleoplasm.”
- o Spelling and grammar in line 340, “... shows that despite frequent mutations are found in Snail1, Lys157 of snial1 is rarely...”
- o Confusing language lines 362-364, “Indeed, HHT is unable to activate JNK nor to upregulate the expression...”

Response 17: Thanks for pointing this out. Accordingly, we have corrected them in the revised manuscript line 257-258, line 386, lines 445-448, respectively.

Point 18:

- Figure legends should include full phrase in addition to abbreviation even if stated in earlier figure legends to help reader understand figure without referencing previous figure legends for clarification.*
- o Figure 3 includes CP, NP, and No, but does describe what they are in the figure legend.
 - o Figure 4I includes CC3 on the Western blot, but no mention of what that is in the figure legend. CC3 is listed in the materials and methods, but is described as caspase-3 in main body text. Some consistency would help the reader.
 - o Figure 5A includes T-JNK on the Western blot, which is not described anywhere in the manuscript. What is the difference between T-JNK and P-JNK? Presumably a typo since the authors are probing for dually phosphorylated p46 and p54. Some explanation in the text would help the reader.

Response 18: Thanks for the constructive suggestions. We have added the full phrase in addition to the abbreviation in the revised Figure legends, respectively.

Point 19:

In the materials and methods section under Western blots, it would be nice to include antibody dilutions used.

Response 19: Thanks for the suggestion. We have added the information of antibody dilutions in the revised materials and methods.

Point 20:

The ruler number markings in Figure 4P are backwards, but maybe that doesn't matter since it's still legible.

Response 20: Thanks for the comment.

Reviewers' Comments:

Reviewer #1:

Remarks to the Author:

The authors have carefully addressed the comments in previous version. The revised manuscript is significantly improved. No further comments for authors.

Reviewer #2:

Remarks to the Author:

The authors have addressed my concerns in a scholarly manner.

Reviewer #3:

Remarks to the Author:

The revised manuscript is dramatically improved and the authors have adequately addressed my concerns with the rigor of experiments (i.e. better quantification of IF data, along with complementation with WT and mutant counterparts).